# Antibacterial potency of type VI amidase effector toxins is dependent on substrate topology and cellular context

Atanas Radkov[1], Anne L Sapiro[1], Sebastian Flores[2], Corey Henderson[3], Hayden Saunders[1], Rachel Kim[4], Steven Massa[5], Samuel Thompson[6], Chase Mateusiak[7], Jacob Biboy[8], Ziyi Zhao[1], Lea M Starita[9,10], William L Hatleberg[11], Waldemar Vollmer[8], Alistair B Russell[12], Jean-Pierre Simorre[13], Spencer Anthony-Cahill[14], Peter Brzovic[15], Beth Hayes[1], Seemay Chou[1]*

[1]Department of Biochemistry and Biophysics, University of California, San Francisco, San Francisco, United States; [2]University of Miami, Miami, United States; [3]InBios International, Seattle, United States; [4]Pacific Northwest University of Health Sciences, Yakima, United States; [5]Department of Biology, Stanford University, Stanford, United States; [6]Department of Bioengineering, Stanford University, Stanford, United States; [7]Computer Science Department, Washington University in St. Louis, St. Louis, United States; [8]Centre for Bacterial Cell Biology, Newcastle University, Newcastle upon Tyne, United Kingdom; [9]Department of Genome Sciences, University of Washington, Seattle, United States; [10]Brotman Baty Institute for Precision Medicine, Seattle, United States; [11]Independent Researcher, Pittsburg, United States; [12]Division of Biological Sciences, University of California, San Diego, La Jolla, United States; [13]Université Grenoble Alpes, Grenoble, France; [14]Chemistry Department, Western Washington University, Bellingham, United States; [15]Department of Biochemistry, University of Washington, Seattle, United States

*For correspondence:
seemay.chou@ucsf.edu

**Abstract** Members of the bacterial *T6SS amidase effector* (Tae) superfamily of toxins are delivered between competing bacteria to degrade cell wall peptidoglycan. Although Taes share a common substrate, they exhibit distinct antimicrobial potency across different competitor species. To investigate the molecular basis governing these differences, we quantitatively defined the functional determinants of Tae1 from *Pseudomonas aeruginosa* PAO1 using a combination of nuclear magnetic resonance and a high-throughput in vivo genetic approach called deep mutational scanning (DMS). As expected, combined analyses confirmed the role of critical residues near the Tae1 catalytic center. Unexpectedly, DMS revealed substantial contributions to enzymatic activity from a much larger, ring-like functional hot spot extending around the entire circumference of the enzyme. Comparative DMS across distinct growth conditions highlighted how functional contribution of different surfaces is highly context-dependent, varying alongside composition of targeted cell walls. These observations suggest that Tae1 engages with the intact cell wall network through a more distributed three-dimensional interaction interface than previously appreciated, providing an explanation for observed differences in antimicrobial potency across divergent Gram-negative competitors. Further binding studies of several Tae1 variants with their cognate immunity protein demonstrate that requirements to maintain protection from Tae activity may be a significant constraint on the mutational landscape of *tae1* toxicity in the wild. In total, our work reveals that Tae diversification has likely been shaped by multiple independent pressures to maintain interactions with binding partners that vary across bacterial species and conditions.

## Editor's evaluation

This study investigates the factors underlying differences in the antimicrobial efficacy of members of the T6SS amidase effector (Tae) superfamily of toxins. This is an interesting and important question from both a physiological and evolutionary perspective.

## Introduction

Bacteria live in dense polymicrobial communities where competition for nutrients and space impacts survival (*Peterson et al., 2020*). To combat neighboring rivals, some Gram-negative bacteria use a contact-dependent toxin delivery system called the type 6 secretion system (T6SS) (*Jani and Cotter, 2010*; *Schwarz et al., 2010*) that injects a cocktail of effectors from the donor cell directly into nearby recipient cells (*Russell et al., 2014a*; *Russell et al., 2011*; *Hood et al., 2010*). A bacterium's T6S arsenals include several types of protein effectors with distinct activities that each compromise essential cellular pathways or components, such as recipient peptidoglycan (PG) or membranes. Although many T6S bacteria deploy similar toxins, interbacterial outcomes can vary considerably depending on the bacterial species engaged in T6S-mediated competition (*Russell et al., 2012*; *Russell et al., 2014b*; *Yu et al., 2021*; *English et al., 2012*; *Zhang et al., 2013*). The molecular basis of this organism-level specificity of interbacterial T6SSs remains poorly understood.

A potential source of specificity could be the T6S-delivered toxins themselves (*LaCourse et al., 2018*). T6S toxins can belong to large superfamilies comprised of diverse homologs. One such example is the T6S *amidase effector* (*tae*) superfamily (*Russell et al., 2012*). The *taes* encode hydrolase enzymes that induce recipient cell lysis by breaking down the major structural component of bacterial cell envelopes, PG (*Russell et al., 2011*). Tae toxins are known to be structurally and biochemically diverse (*Chou et al., 2012*). Given that targeted PG substrates can also vary in a species-dependent manner (*Vollmer et al., 2008a*), such diversity could potentially impact toxin antibacterial specificity. Tae toxins divide into four phylogenetically distinct subfamilies (Tae1–4) (*Russell et al., 2012*) that hydrolyze different amide bonds within PG, and toxins within the same subfamily can have unique PG compositional preferences (*Chou et al., 2012*).

These observations raise the possibility that T6SS organismal specificity may be linked, in part, to the biochemical diversity of delivered toxins. Such a model would predict that physiological features of recipient microbes may have driven divergent evolution of T6S effector diversity. To address this hypothesis, a quantitative, molecular understanding of T6S effector specificity is needed. However, comprehensive functional analyses are challenging for toxins such as the Taes, given that they act on cellular substrates with complex, three-dimensional (3D) structures in vivo. In cells, PG is a cross-linked macromolecule that forms a contiguous, mesh-like network that is structurally heterogeneous, insoluble, and dynamically regulated (*Vollmer and Seligman, 2010*; *Furchtgott et al., 2011*). These features are not well captured through in vitro functional studies that necessarily focus on soluble PG fragments.

To probe Tae–cell wall interactions in a physiologically relevant context, we developed a high-throughput genetic approach known as deep mutational scanning (DMS) (*Furchtgott et al., 2011*) that surveys the entire Tae enzyme for functional determinants of toxicity in live *Escherichia coli* cells. We focused on Tae1 (previously known as Tse1) from the common soil bacterium *Pseudomonas aeruginosa*, which is known to interact with a wide range of microbes, including *E. coli*. We found that an extended surface across the entire Tae1 enzyme is important for mediating in vivo degradation of cell walls. Furthermore, we present evidence that Tae1 functional determinants can be quite distinct depending on the compositional context of PG. Finally, we discovered that the evolutionary potential for Tae1 potency against a broad range of cell wall types may be constrained by orthogonal functional requirements, such as Tae1 interactions with its cognate immunity protein during T6S-mediated competition.

## Results

### DMS identifies determinants of Tae1 function in vivo

To understand how sequence diversity within the Tae protein superfamily impacts interbacterial toxicity, we must first probe how each amino acid residue contributes to Tae function. Biochemical assays of Tae protein binding to and degradation of PG use isolated, purified sacculi (*Chou et al., 2012*), which differ from the turgor-stretched PG layer in the periplasm of living cells. Thus, these assays likely identify only a subset of determinants that govern how Tae engages with PG in vivo. To define comprehensively and quantitatively all the amino acid residues in Tae that are important for its function in vivo, we developed a DMS experiment in live bacterial cells that allows us to examine the contributions of amino acid substitutions at every Tae residue in a physiologically relevant context (*Elazar et al., 2016*; *Barad et al., 2020*).

We adapted a DMS approach (*Fowler and Fields, 2014*; *Rubin et al., 2017*) to study the Tae1 protein from *P. aeruginosa*, which is injected into *E. coli* and leads to cell lysis (*Figure 1A*; *Russell et al., 2011*). We based our screen on a lysis assay that measures the toxic effects of Tae1 on *E. coli* when it is ectopically expressed in the periplasm, which contains the PG (*Chou et al., 2012*). Periplasmic expression of wild-type Tae1 (Tae1^WT), but not catalytically inactive Cys30Ala Tae (Tae1^C30A), results in rapid cell lysis upon induction due to cell wall degradation by the toxin. Cells expressing *tae1*^WT are depleted from the population faster than cells expressing inactive Tae1 (*Figure 1B*). Mutations that increase or decrease the rate of Tae1-induced lysis can be identified by sequencing and analyzing plasmid copy numbers in the population over time.

We generated a barcoded *tae1* plasmid library for massively parallel screening in *E. coli* (*Figure 1C*). Mutations were introduced by error-prone PCR, and library diversity was characterized by sequencing. We focused our analysis on copies of *tae1* that had a single mutation. In total, there were 902 different Tae1 variants, which encompassed at least one amino acid substitution at every position in the 154-residue protein. To ensure Tae1 delivery to the bacterial cell wall, we expressed the variants as fusion proteins with N-terminal periplasm-targeting *pelB* signal sequences. We transformed *E. coli* (BW25113) grown in M9 minimal media with our *tae1* library and sequenced plasmid barcodes associated with each *tae1* gene variant (see 'Materials and methods' for more details), both before and 2 hr after library induction (*Figure 1C*). Tae1-dependent lytic effects were quantified by using the DMS analysis pipeline Enrich2 (*Rubin et al., 2017*), which calculates output-to-input ratios for individual variants, yielding fitness scores that are normalized to wild type (*Figure 1—source data 1*). In our assay, the fitness score reflects a difference in the antibacterial capacity of the Tae1 variant in vivo when compared to the wild type: relative enrichment of a variant indicates reduced Tae1-dependent cell death or loss of function (LOF) and yields a positive fitness score, whereas relative depletion of a variant indicates increased cell death or gain of function (GOF) and yields a negative fitness score.

To validate our approach, we examined the canonical catalytic dyad residues of Tae1 to assess whether our screen accurately identified known functional determinants. Any substitutions of these catalytic residues, Cys30 and His91, should inhibit enzyme activity and lead to LOF. As anticipated, all nonsynonymous amino acid changes at these sites resulted in increased fitness scores, indicating their crucial roles in Tae1 toxicity (*Figure 1C and D*) and confirming that the DMS screen was indeed selecting in vivo functional determinants of Tae PG-degrading toxicity.

### DMS identifies functional hotspots extending around the surface of Tae1

To obtain a global view of the amino acid residues in Tae1 that affect fitness in our screen, we generated an averaged sequence-function map of variant fitness scores (*Figure 2A*). The majority of all unique Tae1 protein variants (54%) were LOF variants with positive fitness scores, whereas only 6% were GOF variants. To determine the functional role of each residue of Tae1, we collapsed and averaged scores across all substitutions at each Tae1 site (*Figure 2A*) and superposed these onto the 3D structure of Tae1 (*Figure 2—figure supplement 1*). For some Tae1 residues, position averages obscured how the degree or direction of functional change depended greatly on the specific amino acid change introduced. Therefore, we used *k*-means clustering, a statistical method for analyzing DMS data (*Thompson et al., 2020*), to classify Tae1 residues in an unbiased way into one of four categories: residues where mutations generally decreased cell lysis (LOF^clus), those where mutations generally increased cell lysis (GOF^clus), those where mutations had variable effects on lysis (mixed), and

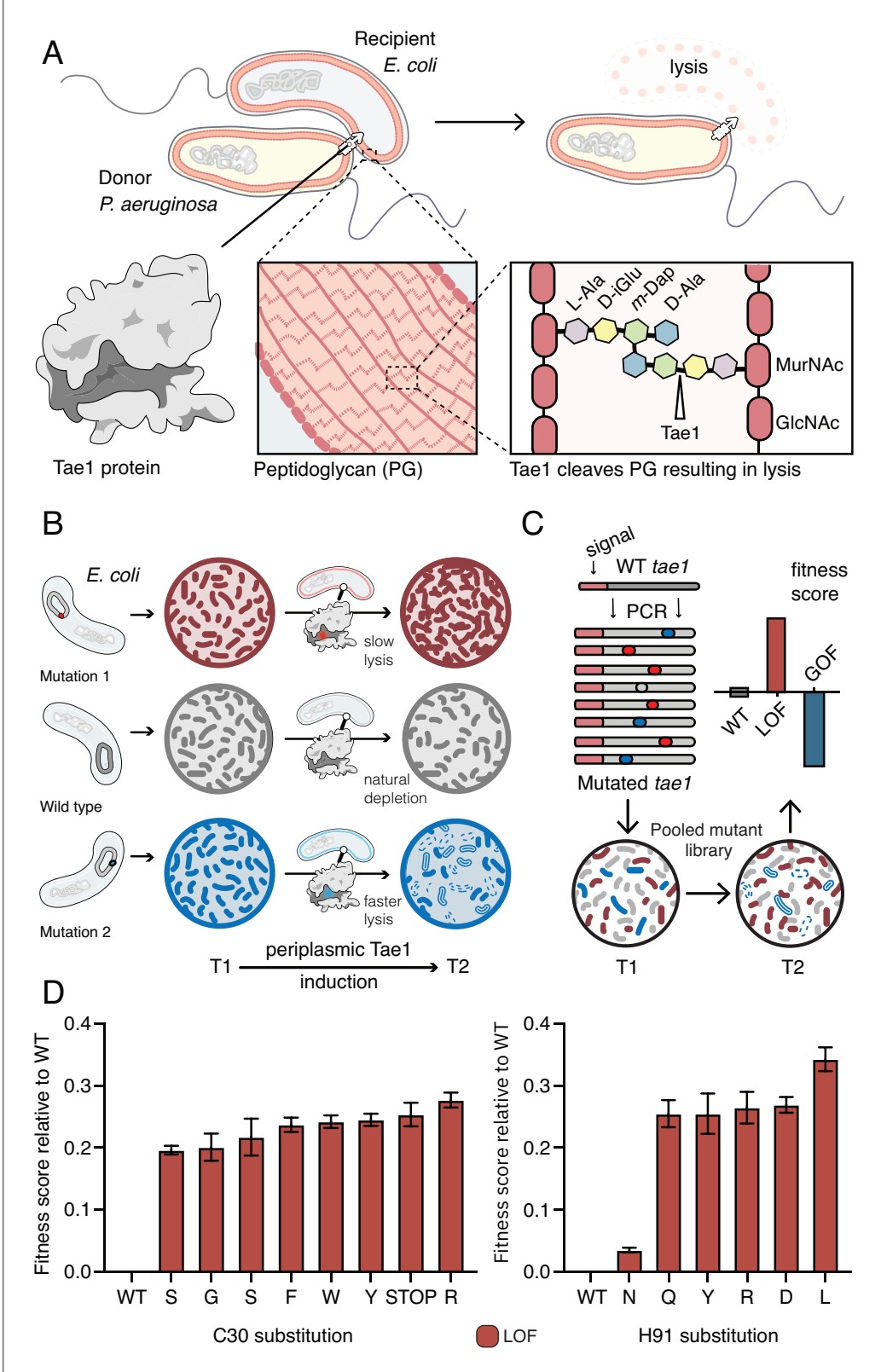

**Figure 1.** Deep mutational scanning (DMS) identifies amino acid residues important for Tae1 function in vivo. (**A**) Illustration of the function of the *P. aeruginosa* Tae1 toxin in lysis of *E. coli*. The donor *P. aeruginosa* cell injects Tae1 into the periplasm of the recipient *E. coli* cell. Tae1 cleaves cell wall peptidoglycan (PG) at the cleavage site, resulting in cell lysis. (**B**) Illustration of lysis assays expressing Tae1 in the periplasm of *E. coli*. When wild-type Tae1

*Figure 1 continued on next page*

*Figure 1 continued*

is expressed, cells are depleted over time (gray). When a loss-of-function mutation such as Tae1[C30A] is expressed, cells grow normally (red). When a gain-of-function mutation is expressed, cells are depleted more quickly over time (blue). T1 = time 1, T2 = time 2. (**C**) Schematic of DMS screen for functional determinants of Tae1. A library of Tae1 mutants was transformed into *E. coli* and periplasmic expression of the mutant toxins was induced. The number of mutant cells remaining at T2 compared to the number at T1 yields a fitness score for each variant. Increased fitness indicates loss-of-function mutations in Tae1, which decrease cell lysis, whereas decreased fitness indicates gain-of-function mutations, which increase lysis. (**D**) DMS-derived relative fitness scores for all substitutions in the library at the Cys30 and His91 positions. Data shown are means ± standard deviation, n = 3. DMS experiment in M9 media was repeated three independent times, each time with at least three technical replicates.

The online version of this article includes the following source data for figure 1:

**Source data 1.** Data for DMS experiments.

those where mutations had little to no effect (tolerant). The vast majority of residues were categorized as either LOF[clus] (56%) or mixed (36%), and fewer were categorized as GOF[clus] (5%) or tolerant (3%) (*Figure 2B*). We mapped these classifications onto the Tae1 structure and found, consistent with other mutational studies, a large majority of buried residues (78%), including the catalytic residues, were categorized as LOF[clus] (*Figure 2C*). Residues on the enzyme surface, by contrast, were a mix of categories, including most of the GOF[clus] residues.

We anticipated that only surface residues within or near the catalytic cleft would show LOF phenotypes. Instead, we found that many surface-exposed residues more distant from the catalytic cleft were also categorized as LOF[clus] (*Figure 2D*). Indeed, a zone of LOF[clus] residues forms a ring around the entire enzyme, suggesting a more distributed set of surfaces may play key roles in cell lysis in vivo than was predicted by in vitro studies (*Chou et al., 2012*). Many of the surface LOF[clus] residues that

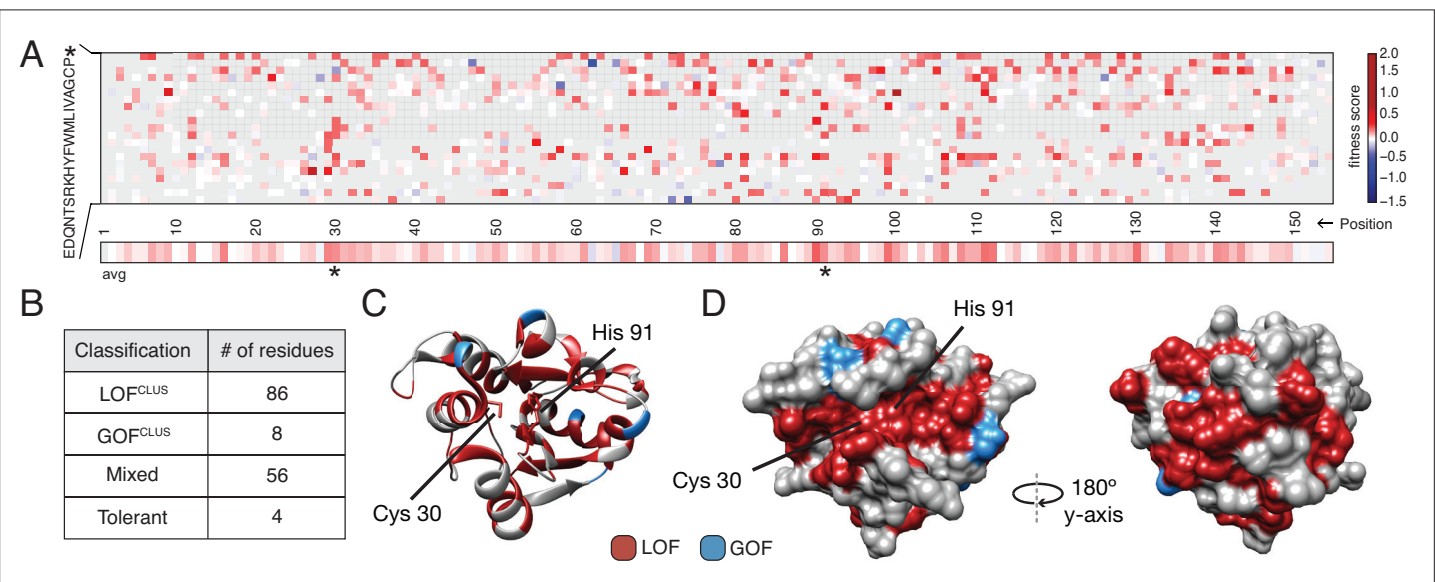

**Figure 2.** Deep mutational scanning (DMS) identifies functional hotspots extending around the surface of Tae1. (**A**) Heatmap representation of fitness scores relative to wild-type Tae1 for all mutations in DMS. Each amino acid position is represented on the x-axis, and each possible amino acid substitution is represented on the y-axis, * indicates stop codon. Each box represents the average fitness score of three replicate experiments. The average fitness score of all substitutions at each residue is depicted across the bottom row. Catalytic residues Cys30 and His91 are indicated with asterisks. Variants in gray were not present in our library. (**B**) Results of *k*-means clustering of DMS data. Based on fitness effects, each residue of Tae1 was labeled loss of function (LOF[CLUS]), gain of function (GOF[CLUS]), mixed, or tolerant. (**C**) Ribbon diagram of Tae1 structure. LOF[CLUS] residues are shown in red, GOF[CLUS] residues are shown in blue, and all other residues are in gray. The catalytic residues of Tae1 (Cys30 and His91) are indicated. Mutations in the enzyme core lead to LOF phenotypes. (**D**) Functional residues on the surface of Tae1. The catalytic surface of the enzyme is shown on the left and the opposite surface, after 180° rotation, is on the right. Color code as in (**C**). LOF[CLUS] mutations are located in large areas of the surface on both sides of the enzyme, and most GOF[CLUS] mutations are also found on the surface. See also *Figure 2—figure supplement 1*.

The online version of this article includes the following figure supplement(s) for figure 2:

**Figure supplement 1.** Average fitness score across each residue.

our screen indicates are crucial for cell lysis in vivo are poorly conserved across the *tae1* superfamily (*English et al., 2012*). We posit that while these mutations are LOF when applied to *E. coli* sacculi, divergent bacterial physiologies in situ may instead drive diversification of this extended surface.

## Tae1 surfaces distal from catalytic core mediate binding and hydrolysis of cell wall network

Our DMS results suggested that surfaces outside the Tae1 catalytic core are functionally important. We observed that several of these distal surface LOF$^{clus}$ sites encoded serine and glycine residues, which have been previously implicated in sugar-binding (*Schanda et al., 2014*; *Maya-Martinez et al., 2018*). Thus, it is possible that the distal LOF$^{clus}$ surfaces are important for interacting with the glycan chains of PG. Alternatively, these distal surfaces could mediate Tae1 interactions with surrounding, untargeted peptide stems within the PG network. To validate that these residues play a role in Tae1 function, we chose nine LOF$^{clus}$ variants distributed across the Tae1 surface (*Figure 3A*) and manually assayed their individual effects through *E. coli* lysis assays (*Figure 3B*, *Figure 3—figure supplement 1*). We focused mainly on serine and glycine residues and introduced the most conservative substitutions represented in the screen (alanines or serines, respectively). Consistent with DMS results, all nine variants showed significantly less lytic activity than Tae1$^{WT}$ (*Figure 3B*).

LOF phenotypes could stem from a variety of mechanisms, including reduced Tae1 protein expression, disruption of protein folding or stability, or direct inhibition of Tae1 PG binding or hydrolysis. To rule out expression and possibly stability-dependent effects, we performed Western blot analyses to measure protein levels of Tae1 variants expressed in *E. coli* (*Figure 3B*). We found that all variants were expressed; five substitutions did not exhibit major visible decreases in protein levels (G42S, S144A, S147A, G48S, G56S). Our results led us to hypothesize that these five surface substitutions may play direct roles in Tae1–PG interactions.

We hypothesized that an extended surface of Tae1 may be required for substrate binding and hydrolysis in vivo. Given that Tae1 likely engages with larger and even sometimes insoluble fragments of PG within the intact cell wall, we conducted a series of in vitro biochemical assays of LOF$^{clus}$ variants using either intact PG sacculi or fragments that encompass more than the minimal peptide region. To address whether distal surfaces of Tae1 bind PG, we experimentally mapped Tae1 binding sites by two-dimensional (2D) nuclear magnetic resonance (NMR) using $^{15}$N-labeled, catalytically inactive Tae1$^{C30A}$ (Tae1$^{N15}$) in complex with larger, soluble fragments of PG. Backbone $^{13}$C, $^{15}$N, and $^1$HN NMR resonances of Tae1 (>89% complete) were assigned using conventional heteronuclear techniques (*Sattler, 1999*). To prepare larger PG fragments for binding experiments, we digested *E. coli* cell wall sacculi using another Tae superfamily member (Tae3 from *Salmonella* Typhi) (*Russell et al., 2012*), which hydrolyzes peptide crosslinks, resulting in long multimeric PG fragments that include both Tae cleavage sites and intact glycan chains (*Figure 3C*) with an average length range of 10 Å to over 300 Å (1 to over 30 dimeric units) (*Harz et al., 1990*).

To map enzyme residues that bind cell wall substrate, we compared NMR spectra of Tae1$^{N15}$ alone or in the presence of digested multimeric PG (*Figure 3—figure supplement 2*). Observed spectral differences allowed us to identify and quantitate chemical shift perturbations (CSPs) across the protein, which serve as indicators of the residues involved in direct substrate interactions (*Schanda et al., 2014*; *Maya-Martinez et al., 2018*). Mapping these shifts onto the Tae1 structure, we observed CSPs for residues located within the catalytic core as well as for a variety of both buried and exposed residues outside of the core (*Figure 3D*). While CSPs for buried residues could be due to interaction-dependent enzyme conformational or dynamical changes (*Gonzalez-Delgado et al., 2020*), contiguous clusters of CSPs localized to the protein surface are suggestive of Tae1–PG binding. Such putative binding regions extended all the way to surfaces on the protein face opposite of the catalytic core. In fact, we identified shifts in six of the validated LOF$^{CLUS}$ residues, Leu21, Gly42, Gly48, Ala52, Gly56, and Ser144 (*Figure 3E*), suggesting that these residues are necessary for Tae1-induced lysis because they bind PG.

To determine if these functional distal residues that bind PG are also required for hydrolysis, we focused our studies on two highly expressed variants, Tae1$^{S144A}$ and Tae1$^{G48S}$, for which we expressed recombinant forms and purified. Size-exclusion chromatography (SEC) analysis of these variants showed similar elution volumes and apparent hydrodynamic radii compared to Tae1$^{WT}$. We then conducted degradative assays with PG sacculi purified from *E. coli*. Tae enzymes were incubated with

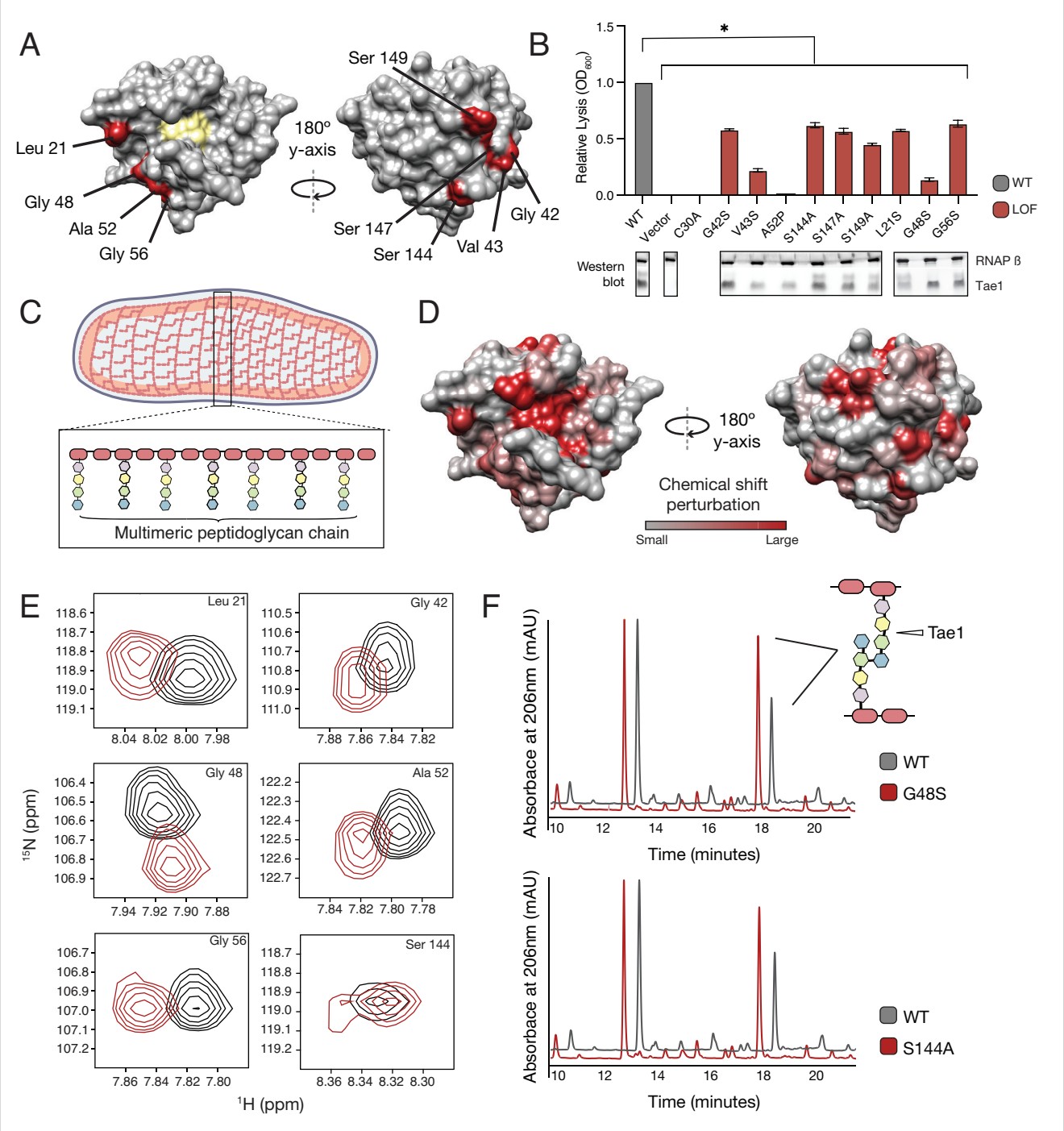

**Figure 3.** Tae1 surfaces distal from catalytic core mediate binding and hydrolysis of cell wall network. (**A**) Nine distal residues on Tae1 that had loss-of-function phenotypes. The catalytic core is highlighted in yellow. (**B**) Lysis assays measuring relative lysis of *E. coli* expressing Tae1^WT, negative controls (vector and C30A), and loss-of-function single amino acid substitutions (G42S, V34S, A52P, S144A, S149A, L21S, G48S, G56S). Data shown are means ± standard deviation, n = 3, * p<0.05. Induced with 0.25% arabinose. Lysis experiment was repeated three independent times, each time with at least three technical replicates. Western blots below show the beta subunit of RNA polymerase (RNAP β) and Tae1 proteins in each assay. All variants and controls showed reduced lysis when compared to wild-type Tae1. (**C**) Schematic of the multimeric glycan chains used as the substrate in nuclear magnetic resonance (NMR) experiments. (**D**) Chemical shift perturbations (CSPs) calculated from NMR spectra of Tae1 with or without multimeric peptidoglycan (PG) chains mapped onto the Tae1 surface structure. Numerous residues on both sides of the Tae1 display CSPs consistent with direct substrate interactions. (**E**) ¹H-¹⁵N Best–Trosy NMR spectra from Tae1 alone (black) and Tae1 incubated with multimeric PG chains (red) for six residues from (**A**) that show CSPs. Residues distal from the Tae1 active site show shifts indicative of interactions with PG. Numerous residues display CSPs consistent with direct substrate interactions. (**F**) High-performance liquid chromatography (HPLC) traces showing degradation of PG in the presence of

*Figure 3 continued on next page*

*Figure 3 continued*

wild-type (black, WT) or mutant (red) Tae1 in vitro. Arrows indicate peaks containing the peptide stem that can be cleaved by Tae1. Mutant Tae1 (S144A and G48S) cleaves PG less well than wild-type Tae1 (29.1 ± 3.2% and 36.4 ± 4.2% less PG., respectively). See also *Figure 3—figure supplements 1 and 2*.

The online version of this article includes the following source data and figure supplement(s) for figure 3:

**Source data 1.** Full images of Western blots depicted in *Figure 3*.

**Figure supplement 1.** Lysis assay growth curves.

**Figure supplement 2.** $^1$H-$^{15}$N Best–Trosy nuclear magnetic resonance (NMR) spectra of Tae1 with and without multimeric peptidoglycan (PG).

PG sacculi, and the resulting fragments were quantitatively assessed using high-performance liquid chromatography (HPLC, *Figure 3F*). Tae1$^{S144A}$ and Tae1$^{G48S}$ cleaved 29.1 ± 3.2% and 36.4 ± 4.2% less PG, respectively, than observed for the wild-type enzyme. Despite being relatively far from the active site, these amino acid substitutions decreased Tae1 activity. Together, these data reveal that numerous sites distributed around the Tae1 surface physically interact with PG, and that these distal interaction sites are important for cell wall-degrading toxicity in vivo. Moreover, these results suggest our DMS-based approach can faithfully uncover novel and unexpected functional determinants in vivo, providing important clues to evolutionary pressures that may be acting on *tae1*.

## The functional landscape of Tae1 is sensitive to PG differences

Our results point to a model in which Tae1 might interact with more structures of PG outside of the catalytic site, and these interactions are important for in vivo toxicity. This model predicts that Tae enzymes are sensitive to differences in cell wall architecture, and broad changes to PG composition would alter Tae1–PG interactions. We hypothesized that such alterations could be captured by comparative DMS analysis across different screening conditions. To address this hypothesis, we conducted a second DMS screen in *E. coli* grown in the presence of D-methionine (D-Met), a condition known for inducing major featural changes in cell wall composition and organization (*Figure 4A*). Specifically, cell walls of *E. coli* grown in the presence of D-Met (*E. coli$^{DMet}$*) have generally reduced PG density and fewer modified PG crosslinks relative to cell walls of *E. coli* grown in the absence of D-Met (*Caparrós et al., 1992*; *Lam et al., 2009*). We verified these trends by HPLC analysis and confirmed that Tae1 maintained its ability to cleave this distinct cell wall type in vitro (*Figure 4—figure supplement 1*).

We performed the same suite of DMS analyses for the second screen as for the first (*Figure 4B*) and found that the calculated fitness landscape shifted dramatically. Approximately 50% fewer residues were categorized as LOF$^{clus}$ (38/86), while the number of mixed residues increased (94/56) as did the number of GOF$^{clus}$ residues (18/8). The number of tolerant residues did not change (four in both) (*Figure 4C*). Overall, 58% of residues were classified differently than in our initial screen in M9 media. We mapped the LOF$^{clus}$ and GOF$^{clus}$ residues onto the Tae1 structure (*Figure 4D*). We also saw that the previous ring-like band of LOF$^{clus}$ determinants emanating from the catalytic groove was greatly diminished in *E. coli$^{DMet}$*. Given that our biochemical experiments suggest that these sites are important for extended PG-binding, we hypothesize that such contact interfaces may be less functionally critical in cellular conditions where the cell wall matrix is more loosely packed around the enzyme. Alternatively, these changes could be indirectly due to other metabolic or physical changes to the cell under D-Met growth conditions. In order to reveal any effects of sacculi architecture on extended Tae1 binding, we tested the two LOF variants Tae1$^{G48S}$ and Tae1$^{S144A}$, confirmed to physically interact with long PG fragments in our NMR experiments, for any differential binding with sacculi obtained from plain or D-Met media. Although we did not observe major binding differences in our pull-down analysis (*Figure 4—figure supplement 2*), we do want to acknowledge the low sensitivity of this assay. There is a need for better tools to improve this analysis. Overall, our comparative screening approach revealed that compositional changes to PG dramatically alter the functional landscape of Tae1, suggesting that these variants are sensitive to differences in cell wall architecture.

The *E. coli$^{DMet}$* DMS screen also yielded a larger and distinct set of top-performing GOF$^{clus}$ variants with amino acid substitutions that localized to both the catalytic groove and opposing faces of Tae1. When we compared fitness scores for the GOF$^{clus}$ variants from the D-Met screen to the initial M9 screen, we found that only a small subset of substitutions showed similar effects across

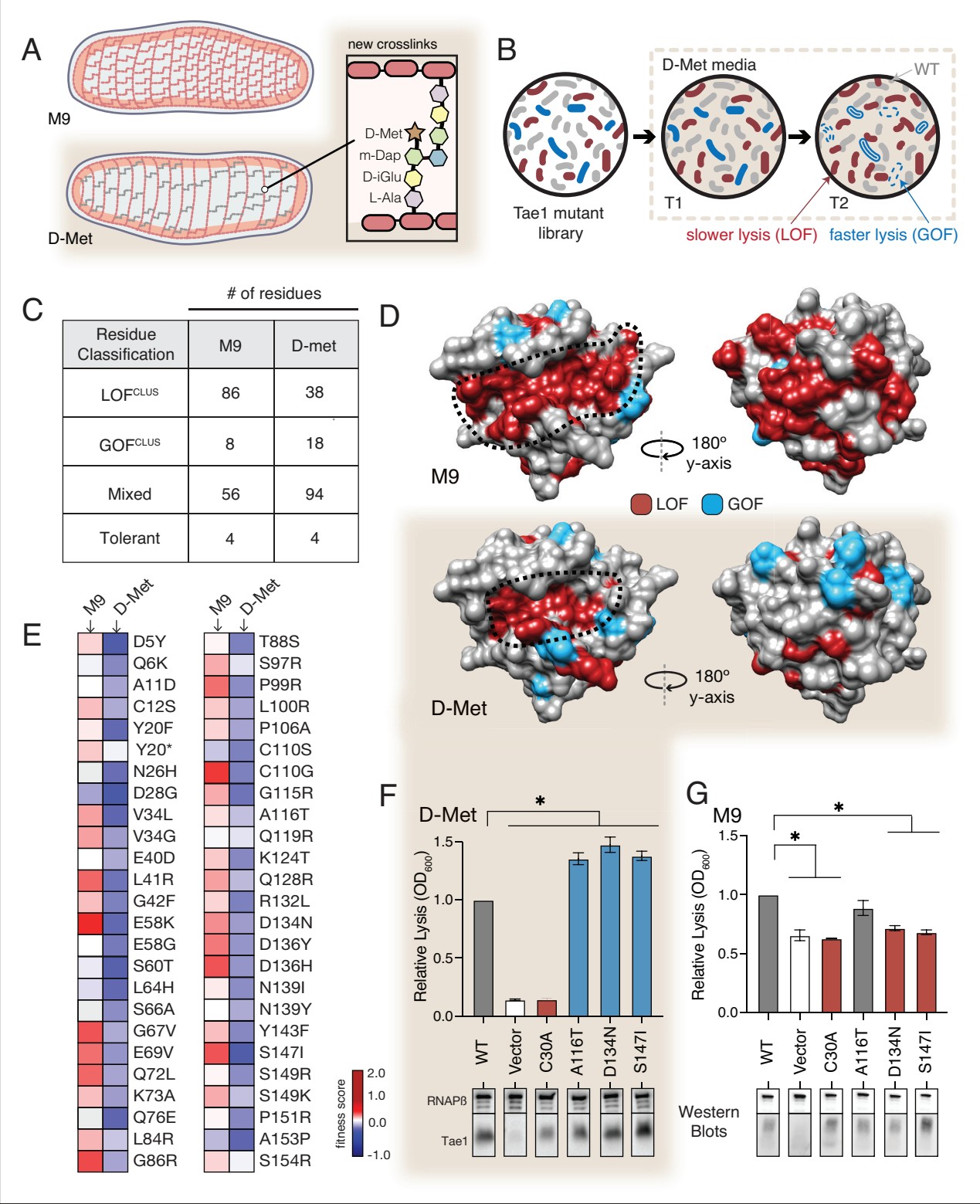

**Figure 4.** The functional landscape of Tae1 is sensitive to cell wall differences. (**A**) Cartoon illustration of cells grown in media containing D-methionine (D-Met) showing altered peptidoglycan (PG) architecture. D-Met becomes incorporated into the peptide stems and there are overall fewer crosslinks in the macromolecule. (**B**) Schematic of the deep mutational scanning (DMS) experiments performed in media containing D-Met. DMS was completed as in *Figure 1* with media containing D-Met. (**C**) Number of residues classified as each of the four cluster types in D-Met DMS screen. The functional classification of residues is dramatically changed from the original screen. (**D**) DMS-derived loss- and gain-of-function (GOF) residues mapped onto the

*Figure 4 continued on next page*

*Figure 4 continued*

surface structure of Tae1 from the initial screen (left) and the D-Met screen (right). Compared with regular media, there were fewer loss-of-function and more gain-of-function residues in the presence of D-Met on both the catalytic face and back of the enzyme. (**E**) Heatmap representing fitness scores for the top 50 GOF individual amino acid substitutions in D-Met and M9 media. n = 3. DMS in D-Met media was repeated three independent times, each time with at least three technical replicates. The majority of the GOF substitutions in D-Met are condition specific. (**F**) Lysis assays with wild-type Tae1, empty vector and catalytically inactive Tae1^C30A as negative controls, and three GOF Tae1 variants (A116T, D134N, S147I). Data are means ± standard deviation, n = 3, *p<0.05. Induced with 0.0025% arabinose. Lysis experiment was repeated three independent times, each time with at least three technical replicates. Western blots below measure Tae1 protein expression with RNAP as a control. All three GOFs from DMS show increased lysis phenotype in individual lysis assays. (**G**) Lysis assays with the same Tae1 variants as in (**F**) but in M9 media. Western blots below measure Tae1 protein expression with RNAP as a control. The three D-Met GOF substitutions are condition specific. See also *Figure 4—figure supplements 1 and 2*.

The online version of this article includes the following source data and figure supplement(s) for figure 4:

**Source data 1.** Full images of Western blots depicted in *Figure 4*.

**Figure supplement 1.** D-methionine (D-met) alters density and type of crosslinks.

**Figure supplement 2.** Sacculi architecture does not impact interactions between sacculi and Tae1 enzyme variants.

**Figure supplement 2—source data 1.** Full images of Western blots depicted in *Figure 4—figure supplement 2*.

both conditions (*Figure 4E*). Condition-specific phenotypes suggest that Tae1 toxicity in vivo is highly context-dependent and optimizable for specific PG forms. To test this hypothesis, we conducted lysis assays in D-Met and M9 media for three GOF variants uniquely identified in the D-Met screen (*Figure 4F and G*). All three substitutions increased cell lysis in *E. coli* grown in D-Met but not M9 minimal media. For two of the variants, Asp134Asn and Ser147Ile, the increase in lysis may be due to higher expression levels based on Western blot analysis (*Figure 4F and G*). Together, these results suggest that Tae enzymes could possibly be engineered to target specific bacteria with certain cell wall structures. Given that the structure and composition of PG vary across bacterial species, our results also potentially point to a selective pressure underlying natural sequence-based diversity in the Tae1 superfamily. Namely, cell wall features associated with particular rival species a given bacterium must antagonize may drive evolved substrate specificity differences in interbacterial competition toxins such as the Tae enzymes.

## A hyperactive variant of Tae1 is naturally encoded by other *tae* homologs

A number of context-independent Tae1 variants that significantly outperformed wild-type enzyme across both DMS conditions were also identified through our comparative approach. We were particularly struck by one variant, Tae1^C110S, which is modified at a partially buried residue sitting directly behind the catalytic Cys30 site (*Figure 5A*). The Cys110Ser substitution was the top-scoring GOF variant in the D-Met screen and ranked 11th amongst GOF variants in the M9 screen. Using the *E. coli* lysis assay, we first confirmed that Tae1^C110S indeed lysed cells at a faster rate than Tae1^WT in cells grown in both D-Met and M9 minimal media (*Figure 5B*). To rule out indirect effects on toxicity caused by increased protein levels, we assessed Tae1^C110S expression by Western blot analysis and did not observe any increase relative to Tae1^WT (*Figure 5B*).

The observation that Tae1 activity could generally be tuned for greater toxicity led us to ask why *tae1* from *P. aeruginosa* does not naturally encode Ser110. We considered several explanations, none of which are mutually exclusive. First, it is possible that structural rearrangements introduced by the Cys110Ser substitution may disrupt Tae1 interactions with critical molecular partners present in natural T6S-mediated interbacterial interactions but not in our DMS set-up. During T6S competition, Tae1 from *P. aeruginosa* must be delivered to neighboring cells by a complex T6S secretion apparatus that we bypassed with an engineered secretion signal in our screen. Furthermore, Tae1 injected into nearby kin cells is deactivated through binding by a cognate immunity protein, *T6S amidase immunity protein 1* (Tai1), preventing lysis. Tai1 is specifically encoded by *P. aeruginosa* and not *E. coli*. Second, it is possible that we have not sampled Tae1 fitness across contexts that accurately represent bacterial rivals or competition conditions most relevant for *P. aeruginosa* in the wild.

We therefore investigated whether Ser110 naturally occurs in any Tae1 representatives outside of *P. aeruginosa* by searching for family members that encode a serine at residue positions comparable to Cys110, as predicted by structure-based alignments. Surprisingly, we found that approximately half of

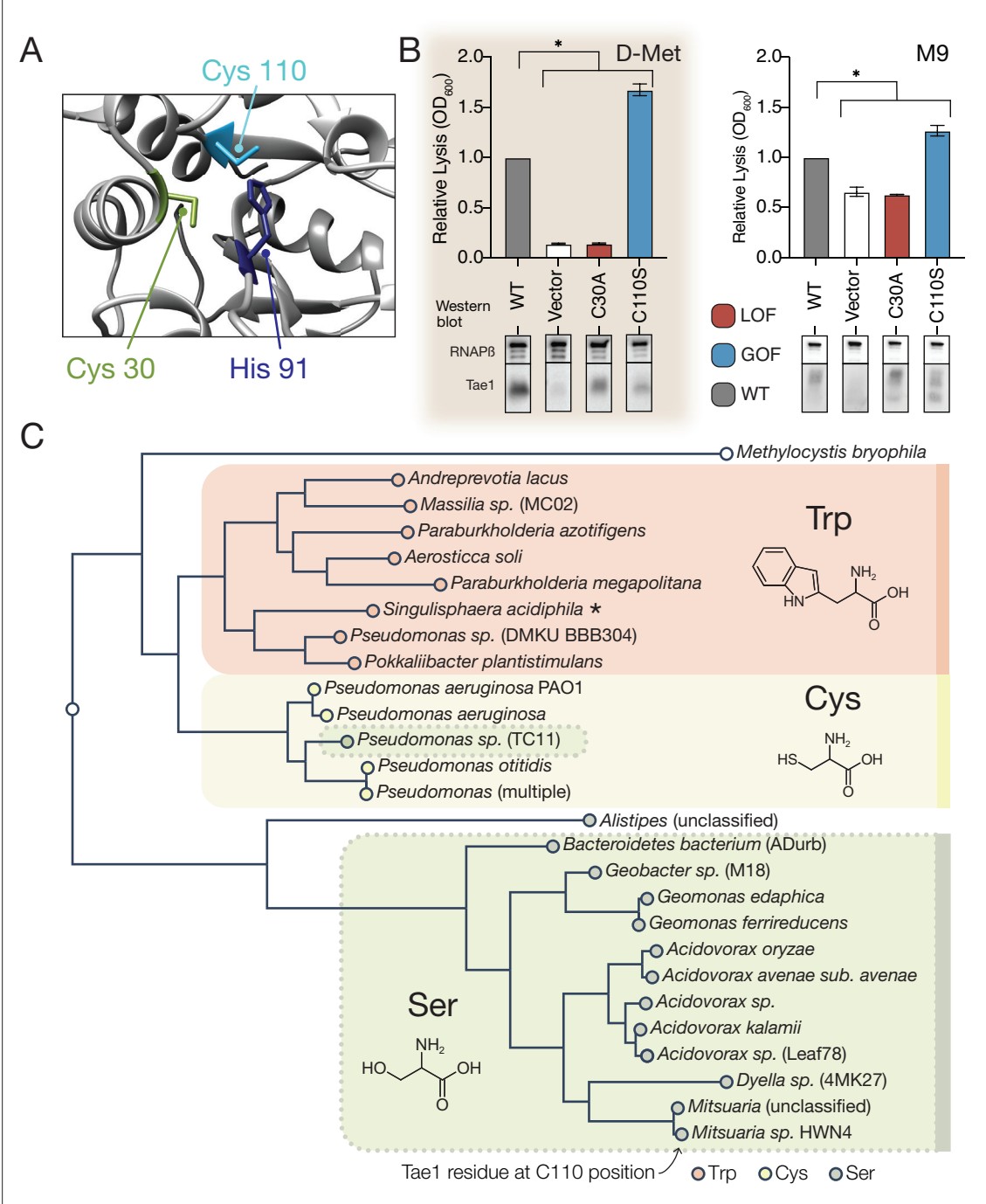

**Figure 5.** A hyperactive variant of Tae1 is naturally encoded by other *tae* homologs. (**A**) Ribbon structure diagram at the Tae1 catalytic site. Catalytic dyad residues Cys30 and His91 are indicated along with gain-of-function (GOF) Cys110. (**B**) Lysis assays with wild-type Tae1, empty vector and catalytically inactive Tae1$^{C30A}$ as negative controls, and GOF variant Tae1$^{C110S}$ in D-methionine (D-Met) and M9 media. Data are means ± standard deviation, n = 3, *p<0.05. Induced with 0.0025% arabinose. Lysis experiment was repeated three independent times, each time with at least three technical replicates. Western blots below measure Tae1 protein expression with RNAP as a control. The C110S GOF substitution is not sensitive to the D-Met-induced changes in cell wall architecture. This is part of the same blot shown in *Figure 4F*. (**C**) Tae1 family phylogenetic tree. Species with Ser at position 110 are indicated in blue, while those with Cys are in black and those with Trp are in purple. Some but not all Tae1 proteins have evolved to include a Ser at position 110.

The online version of this article includes the following source data for figure 5:

**Source data 1.** Full images of Western blots depicted in *Figure 5*.

**Source data 2.** Tae1 protein sequences used in the phylogenetic tree in *Figure 5*.

*tae1*-carrying bacterial species encoded serine residues at the Cys110 position, while some encoded tryptophan (*Figure 5C*). To examine how these three variant classes are taxonomically distributed, we generated a phylogenetic tree of all associated bacterial species and observed correlated divisions across taxonomic clades and the residues at the Cys110 position, with the exception of one *Pseudomonas* species (*Figure 5C*). Considering this observation, we hypothesized that *tae1* evolution may have been locally constrained due to the requirement for Tai1-dependent immunity to prevent self-antagonism.

## Hyperactive variant Tae1$^{C110S}$ evades binding and inhibition by cognate immunity

We reasoned that if Tae1 proteins are too structurally divergent, the toxin may evade binding by the cognate Tai1 partner in *P. aeruginosa* (Tai1$^{PA}$) and consequently antagonize kin cells or other closely related species. To assess whether the Tae1$^{C110S}$ variant affects Tae1–Tai1 interactions, we first assessed whether Tae1$^{C110S}$ adopted any major structural changes by solving its 3D crystal structure (PDB ID: 7TVH) (*Table 1*). We did not observe any major structural rearrangements that would immediately implicate distinct modes of Tai1$^{PA}$-binding (RMSD = 0.746, *Figure 6A*). However, we did observe some key side chain alterations near the Tae1$^{C110S}$ active site, which is the enzyme region most intimately engaged with Tai1$^{PA}$ upon binding (*Figure 6—figure supplement 1*). Most notably, we found that His91 protruded away from the active site of Tae1$^{C110S}$, which would sterically clash with a loop of Tai1$^{PA}$ that typically inserts into the catalytic cleft (*Figure 6B*). Our structural observations are consistent with the hypothesis that the Cys110 position is important for Tai1 recognition; therefore, serine at this position in *P. aeruginosa* may be disfavored to preserve immunity to the toxin. We generated a tree for the Tai1 protein family, similar to our tree for the Tae1 protein family. Consistent with this possibility, the tree for Tai1 has a similar branching pattern that is coincident with the residue variation at this position (*Figure 6—figure supplement 2*).

We tested our prediction through a series of quantitative in vitro Tae1–Tai1 binding experiments. We measured Tai1-binding affinity (Kd) for recombinant Tae1$^{WT}$ and Tae1$^{C110S}$ proteins by isothermal titration calorimetry and found that Tae1$^{WT}$ bound Tai1 with ~36-fold greater affinity than Tae1$^{C110S}$ (9.5 nM Kd and 349.6 nM Kd, respectively) (*Figure 6C*). Toxin–immunity complexes are typically associated with binding affinities in the nM range; thus, reduction may preclude the ability of Tai1 to effectively bind its toxin partner in vivo. To test whether Tae1$^{C110S}$ evades Tai1-dependent immunity in vivo, we constructed a strain of *P. aeruginosa* harboring this mutation (*PA$^{C110S}$*) with a wild-type strain (*PA$^{WT}$*). We reasoned that if Tae1$^{C110S}$ escapes Tai1-binding and therefore immune protection, we would observe a decrease in *PA$^{C110S}$* fitness compared to a wild-type strain (*PA$^{WT}$*) specifically when grown on solid media which induces T6S (*Figure 6D and E*). Indeed, for *PA$^{C110S}$* we observed substantially reduced growth compared to *PA$^{WT}$* on solid media (*Figure 6F*) but not in liquid media, where T6S is not induced (*Figure 6—figure supplement 3*). This phenotype strongly suggests that Tae1$^{C110S}$ lyses kin cells at a faster rate than Tai1 can bind and inhibit killing, leading to a fitness cost for this strain. Furthermore, we conducted a contact-dependent competition. This assay was done in the same way as the fitness assays but with the addition of a prey cell *E. coli* (*Figure 6G and H*). Here, an attacking *PA* strain could still inject Tae1 into kin cells but also into the prey cells. Again, for *PA$^{C110S}$* we observed a substantially reduced competitive index compared to *PA$^{WT}$* (*Figure 6I*). This observation further supports that the *PA$^{C110S}$* strain likely lacks sufficient Tai1 immunity, in turn leading to a diminished competitive capacity against *E. coli* prey cells.

Together, our in vitro and in vivo results suggest that the fitness cost of evading Tai1-binding likely limits the proliferation of Tae1$^{C110S}$ in the wild. This interaction presents an evolutionary bottleneck that impedes certain toxin variants, such as Tae1$^{C110S}$, from broadly taking hold across all species. More generally, our DMS screening approach allowed us to systematically probe function across different in vivo contexts, identifying two distinct but important selective pressures – substrate-specificity and toxin–immunity interactions – that may, in combination, shape diversity across the Tae1 toxin superfamily.

**Table 1.** Primers used in this study.

| Name | Sequence |
| --- | --- |
| pET22_Tae1_C110S_F1 | CCGATGTGCTGGTCCGGCAGCATCGCCGGC |
| pET22_Tae1_C110S_F2 | GGTGGTGGTGCTCGAG |
| pET22_Tae1_C110S_R1 | GCCGGCGATGCTGCCGGACCAGCACATCGG |
| pET22_Tae1_C110S_R2 | TGGATATCGGAATTAATTCGGATC |
| pET29_Tae1_C110S_F1 | CCGATGTGCTGGTCCGGCAGCATCGCCGGC |
| pET29_Tae1_C110S_F2 | GGTGGTGGTGCTCGAG |
| pET29_Tae1_C110S_R1 | GCCGGCGATGCTGCCGGACCAGCACATCGG |
| pET29_Tae1_C110S_R2 | GGGAATTCCATATGGACAGTCTCGATCAATGC |
| pBAD_Tae1_F | TAGTACAGAGAATTCACCATGAAATACCTGCTGCCGACCGCTGCTGC |
| pBAD_Tae1_R | TCAATCAGTATCTAGATTACTCGAGACTGGCCCTGGGCAGGCTG |
| pEXG2_pppA_F1 | TGTTAAGCTAAAGCTTGTCGGTCGCTATTTCCCGCTGAC |
| pEXG2_pppA_F2 | GCATACCCGGTTAAGGACAACTGATGGTGAACTCGAGCCGCAAGCATGCTGAA |
| pEXG2_pppA_R1 | TCAATCAGTATCTAGAGCCGGTGAGGATCTCGTAGAGCAC |
| pEXG2_pppA_R2 | CCGGAAGTTCTATGTCCATGTCCGTCTCAGAATTCAGCATGCTTGCGGCTCGAGTT |
| pEXG2_retS_F1 | TCAATCAGTATCTAGACGGAAACCCAGCCGATCATGG |
| pEXG2_retS_F2 | AACTCGAGCCGCAAGCATGCTGAAGGCGAAGTCCCTTCGAAGGTG |
| pEXG2_retS_R1 | TGTTAAGCTAAAGCTTCCAGCATCTTCAGGTAGACGC |
| pEXG2_retS_R2 | TTCAGCATGCTTGCGGCTCGAGTTGCCTACTGGGTCGGCGAACGC |
| pEXG2_Tae1C110S_F1 | TCAAGTACTAGAGCTCGTGGTGCACCGCGAGGACATCTC |
| pEXG2_Tae1C110S_F2 | CTGCCACTCCAGCACATCGGGTAC |
| pEXG2_Tae1C110S_R1 | TATCAGAAACCTGCAGGTAGGCCAGCGACTGCATGCCGTAG |
| pEXG2_Tae1C110S_R2 | GATGTGCTGGAGTGGCAGCATCGCC |
| pBAD_Tae1L21S_F | AGCTGGGACAAGAGCTACAGCGCCGGCACCCCG |
| pBAD_Tae1L21S_R | CTTGTTCGGGGTGCCGGCGCTGTAGCTCTTGTC |
| pBAD_Tae1G48S_F | GGCGTACCGATGCCCCGCAGCAACGCCAACGCC |
| pBAD_Tae1G48S_R | GACCATGGCGTTGGCGTTGCTGCGGGGCATCGG |
| pBAD_Tae1G56S_F | GCCAACGCCATGGTCGACAGCCTGGAGCAGAGC |
| pBAD_Tae1G56S_R | GGTCCAGCTCTGCTCCAGGCTGTCGACCATGGC |
| pBAD_Tae1G42S_F | TCGGTGGCCGCCGAGCTGAGCGTACCGATGCCC |
| pBAD_Tae1G42S_R | GCCGCGGGGCATCGGTACGCTCAGCTCGGCGGC |
| pBAD_Tae1V43S_F | GTGGCCGCCGAGCTGGGCAGCCCGATGCCCCGC |
| pBAD_Tae1V43S_R | GTTGCCGCGGGGCATCGGGCTGCCCAGCTCGGC |
| pBAD_Tae1A52P_F | CCCCGCGGCAACGCCAACCCGATGGTCGACGGC |
| pBAD_Tae1A52P_R | CTCCAGGCCGTCGACCATCGGGTTGGCGTTGCC |
| pBAD_Tae1S144A_F | CTCAACTACTACGTCTACGCCCTGGCCAGTTGC |
| pBAD_Tae1S144A_R | CAGGCTGCAACTGGCCAGGGCGTAGACGTAGTA |
| pBAD_Tae1S147A_F | TACGTCTACTCCCTGGCCGCCTGCAGCCTGCCC |
| pBAD_Tae1S147A_R | GGCCCTGGGCAGGCTGCAGGCGGCCAGGGAGTA |
| pBAD_Tae1S149A_F | TACTCCCTGGCCAGTTGCGCCCTGCCCAGGGCC |

*Table 1 continued on next page*

*Table 1 continued*

| Name | Sequence |
| --- | --- |
| pBAD_Tae1S149A_R | GAGACTGGCCCTGGGCAGGGCGCAACTGGCCAG |
| pBAD_Tae1A116T_F | TGCGGCAGCATCGCCGGCACTGTCGGCCAGAGC |
| pBAD_Tae1A116T_R | GCCCTGGCTCTGGCCGACAGTGCCGGCGATGCT |
| pBAD_Tae1D134N_F | CAGGTGTGGAATCGCACCAACCGCGACCGCCTC |
| pBAD_Tae1D134N_R | GTAGTTGAGGCGGTCGCGGTTGGTGCGATTCCA |
| pBAD_Tae1S147I_F | TACGTCTACTCCCTGGCCATTTGCAGCCTGCCC |
| pBAD_Tae1S147I_R | GGCCCTGGGCAGGCTGCAAATGGCCAGGGAGTA |
| DMS_Lib_Amplify_F | AATGATACGGCGACCACCGAGATCTACACCTTTCGGGCTTTGTTAGCAG |
| DMS_Lib_Amplify_R1 | CAAGCAGAAGACGGCATACGAGATGGAACGGAGTGACTGGAGTTCAGACGTGTGCTCTTCCGATCTTCGATAGTAGCCTATGAAGCATGCTTACTA |
| DMS_Lib_Amplify_R2 | CAAGCAGAAGACGGCATACGAGATCGGCACATGTGACTGGAGTTCAGACGTGTGCTCTTCCGATCTTCGATAGTAGCCTATGAAGCATGCTTACTA |
| DMS_Lib_Amplify_R3 | CAAGCAGAAGACGGCATACGAGATTACCCTTGGTGACTGGAGTTCAGACGTGTGCTCTTCCGATCTTCGATAGTAGCCTATGAAGCATGCTTACTA |
| DMS_Lib_Amplify_R4 | CAAGCAGAAGACGGCATACGAGATTAGGTGTCGTGACTGGAGTTCAGACGTGTGCTCTTCCGATCTTCGATAGTAGCCTATGAAGCATGCTTACTA |
| DMS_Lib_Amplify_R5 | CAAGCAGAAGACGGCATACGAGATGAATCACTGTGACTGGAGTTCAGACGTGTGCTCTTCCGATCTTCGATAGTAGCCTATGAAGCATGCTTACTA |
| DMS_Lib_Amplify_R6 | CAAGCAGAAGACGGCATACGAGATCATAGAGTGTGACTGGAGTTCAGACGTGTGCTCTTCCGATCTTCGATAGTAGCCTATGAAGCATGCTTACTA |
| DMS_custom_seq_F | GGTGGTGCTCGACGATAGCACGTCTCTGACACCGAATTC |
| DMS_custom_seq_R | GCCCAGCCGGCGATGGCCATGGATATCGGAATTAATTC |

## Discussion

The Tae superfamily of interbacterial toxins is phylogenetically, structurally, and functionally diverse (*Russell et al., 2012*; *Chou et al., 2012*). Here, we show that sequence-based differences in *tae* homologs shape Tae proteins and their complex substrate, the PG that constitutes the bacterial cell wall. Our studies revealed that Tae1 engages PG through a circumferential interface that is more extensive than most proteases or other enzymes. This unexpectedly large interaction surface may be key for dictating species-specific toxicity of Tae proteins and consequently serve as a major driver of Tae superfamily diversification. This surface could also mediate processivity across PG. Taes have many other binding partners as well. These toxins interact with the T6SS apparatus for secretion, which likely limits effector size to a single, minimal catalytic unit (*Russell et al., 2014a*; *Basler et al., 2012*). Therefore, unlike other larger multi-domain housekeeping PG hydrolases (*Vollmer et al., 2008b*), Tae toxins rely on a relatively restricted set of surfaces to mediate many distinct, functional interactions. Our unbiased, high-throughput DMS analyses revealed that determinants for Tae1 cell wall-binding and immunity-binding partially overlap. Our findings suggest that accommodation of multiple, distinct functions likely constrains evolution of Tae enzymes in nature despite intrinsic potential for enhanced toxicity. Extensive Tae functional surfaces may also point to the existence of additional unknown interaction partners within bacteria. This principle may prove true for other T6S toxins and proteins; comprehensive mapping of functional determinants by DMS may power further functional discovery.

Complementing traditional biophysical approaches with DMS allowed us to probe molecular interactions in physiological states that are difficult to capture in vitro. Our current understanding of enzyme interactions with the bacterial cell wall draws mostly from biochemical analyses conducted in solution, posing a fundamental challenge for proteins with 3D highly crosslinked macromolecular substrates. Considering the architectural complexity of the bacterial cell wall, a large ring-like surface looping around Tae1 seems critical for toxicity in vivo. This observation could be due to at least two possible modes of binding that we posit may be unique to enzymes with macromolecular substrates.

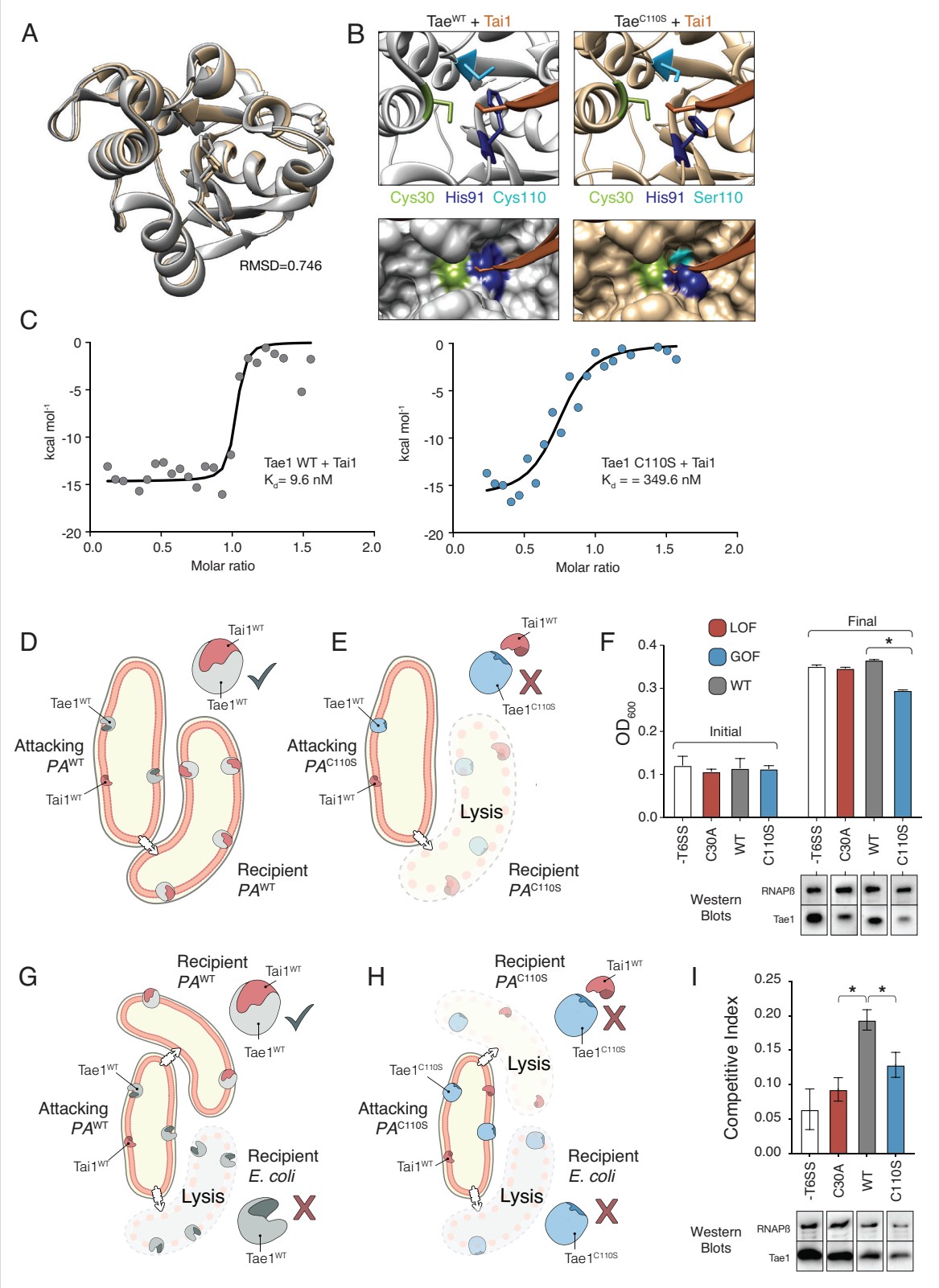

**Figure 6.** Hyperactive variant Tae1C110S evades binding and inhibition by cognate immunity. (**A**) Tae1WT ribbon structure diagram (silver) and Tae1C110S ribbon structure diagram (blue) with the His91 side chain shown. C110S mutation does not cause major structural changes, but alters side chains in the active site. (**B**) Ribbon diagrams of the active sites of Tae1WT (left) and Tae1C110S (right) in complex with Tai1 (pink). Side chains of Cys30, His91, and Cys110 are indicated in gray, and Ser110 in blue. His91 clashes with Tai1 in the active site. (**C**) Isothermal titration calorimetry determination of binding

*Figure 6 continued on next page*

*Figure 6 continued*

constants between wild-type Tae1 and Tai1 (left) and Tae1$^{C110S}$ and Tai1 (right). Tae1$^{C110S}$ binds poorly to Tai1. (**D, E**) Schematic depicting Tae1 and Tai1 interactions in wild-type *P. aeruginosa* and Tae1$^{C110S}$ *P. aeruginosa*. When wild-type Tae1 is injected into a neighboring cell of the same species, Tai1 binds to it, conferring immunity. Tai1 cannot bind Tae1$^{C110S}$, thus the mutant Tae1 can cause cell lysis. (**F**) Fitness assays of *P. aeruginosa* expressing wild-type Tae1 or Tae1$^{C110S}$ from the endogenous locus. *P. aeruginosa* lacking a type 6 secretion system (-T6SS) or expressing catalytically inactive Tae1 (C30A) were used as controls. Data are means ± standard deviation, n = 3, *p<0.05. Fitness assays were repeated three independent times, each time with at least three technical replicates. Cells expressing Tae1$^{C110S}$ have decreased fitness when compared with wild-type Tae1. See also ***Figure 6— figure supplements 1–3***. (**G, H**) Schematic depicting contact-dependent competition between *E. coli* and either wild-type *P. aeruginosa* or Tae1$^{C110S}$ *P. aeruginosa*. The key point here is that an attacking *P. aeruginosa* cell may inject Tae1 effector into a kin cell or into a prey *E. coli* cell. (**I**) Competition assays of *P. aeruginosa* expressing wild-type Tae1 or Tae1$^{C110S}$ from the endogenous locus. *P. aeruginosa* lacking a -T6SS or expressing catalytically inactive Tae1 (C30A) were used as controls. Data are means ± standard deviation, n = 3, *p<0.05. Competition assays were repeated three independent times, each time with at least three technical replicates. Cells expressing Tae1$^{C110S}$ have a decreased competitive index when compared with wild-type Tae1.

The online version of this article includes the following source data and figure supplement(s) for figure 6:

**Source data 1.** Full images of Western blots depicted in ***Figure 6***.

**Figure supplement 1.** Tae1–Tai1 interactions.

**Figure supplement 2.** Tai1 family phylogenetic tree.

**Figure supplement 2—source data 1.** Tai1 protein sequences used in the phylogenetic tree in ***Figure 6—figure supplement 2***.

**Figure supplement 3.** Competition assay in liquid culture.

First, we theorize that a more extensive, nonlinear substrate binding domain may enable an enzyme to specifically recognize a particular conformational state. 3D configurations of the cell wall may be an important feature for a toxin like Tae1 to detect, given that this could affect protein localization to specific cell wall features that have distinct shapes. It has been previously observed that a critical hole size in the cell wall is necessary for bacterial lysis ***Daly et al., 2011***; thus localized action of PG enzymes may provide an important advantage for Tae toxins. Another potential advantage of binding a specific substrate configuration is that this would enable a Tae enzyme to distinguish between different species or incorporation states of PG fragments. Indeed, certain endogenous enzymes, such as FtsN, interact exclusively with complex PG chains of a certain length (***Ursinus et al., 2004***).

Another possibility is that the large functional surface comprises several different PG-binding domains that streamline substrate recognition or underlie enzyme processivity. This has been suggested for a cell wall digesting enzyme, Lysostaphin from *Staphylococcus*, which exhibits two large PG-binding surfaces on opposite sides of the enzyme (***Gonzalez-Delgado et al., 2020***). Multi-step binding would support the 'smart autolysins' model postulated by ***Koch, 1990***. This model hints at the ability of hydrolase enzymes to recognize 'stretched' areas of PG. The resulting advantage for hydrolytic cell wall enzymes is that this type of recognition decreases substrate search dimensionality from random 3D diffusion to facilitated diffusion by searching along the PG sacculus one plane at a time (***Kari et al., 2017***). This may be particularly important for rapid, effective recognition of substrates within a large cage-like 3D target like the cell wall, which is key for potent, fast-acting toxins. This mechanism is also commonly adopted by DNA-binding proteins, which dock nonspecifically to DNA before searching for specific cut sites (***Terry et al., 1985***; ***Stracy et al., 2021***). Multi-step binding may also lead to processive activity along PG by ensuring that Tae enzymes remain tethered to the cell wall substrate after each hydrolysis event, which would allow for 'stepwise chewing.' This would be yet another mechanism to increase the odds of localized, lytic activity.

In total, our combined use of biophysical and high-throughput genetic approaches uncovered novel insights into Tae toxin activity inside bacterial cells, providing a window into how small, single-domain enzymes carry out several, distinct functions in vivo. Our leveraging of a multipronged strategy was critical for overcoming technical obstacles inherent to proteins that interact with complex, insoluble molecules, providing access to a class of macromolecular interactions that are not well understood. The biophysical and kinetic parameters of proteins with 3D highly crosslinked macromolecular substrates (***Broendum et al., 2021***; ***McLaren and Packer, 1970***) are dissimilar from those well-established for small, soluble substrates, and future adoption of combinatorial approaches such as ours may improve our understanding of general binding principles across these important enzymes that are ubiquitous across life. Additionally, the ability to quickly generate rich quantitative heatmaps for enzymes across different experimental conditions enables more sophisticated computational analyses. Predictive

representation learning of unbiased, functional datasets across multiple biological conditions could help us anticipate a priori the functional consequences of sequence-based phylogenetic diversity. A deeper understanding of functional design principles would also enable more effective leveraging of evolutionary innovations to engineer new molecules tailored to physiological contexts with important, useful applications.

## Materials and methods
### Preparation of long chains of PG

PG isolation and HPLC analysis were conducted according to an established protocol (*Stankeviciute et al., 2019*) with the following modifications. Briefly, *E. coli* BW25113 cells were grown in 200 ml LB media to mid-log phase. The cells were harvested by centrifugation at 5000 × *g* for 15 min, after which they were suspended in 300 μl of fresh LB media and added to 600 μl 6% SDS solution in a 2 ml Eppendorf tube. The cells were boiled for 3 hr in a heat block and then maintained in the SDS solution at 4°C until needed. The boiled sacculi were washed five times with MQ water to remove the SDS. Each wash consisted of 15 min centrifugation at 21,000 × *g* followed by decanting of the supernatant and resuspending in 1 ml of MQ water by vortexing. After the final wash, the sacculi were suspended in Pronase E digestion buffer (10 mM Tris-HCl pH 7.2% and 0.06% NaCl) and were treated for 2 hr at 60°C with activated Pronase E (100 μg/ml final concentration, pretreated for 2 hr at 60°C in the digestion buffer; supplier: VWR). The sacculi were incubated at 100°C for 10 min to inactivate the Pronase E and were washed three times with 1 ml of MQ water, with vortexing to resuspend. The sacculi were treated with purified Tae3 enzyme (10 μM) overnight at 37°C in 20 mM HEPES pH 7.0 and 100 mM NaCl buffer. The PG fragments were finally dialyzed using 500 Da molecular weight cutoff tubing (Biotech CE Tubing, Spectra/Por) and dried on a Vacufuge concentrator to remove all liquid. They were stored at –80°C and were redissolved before each NMR experiment with minimal volume to yield the desired enzyme to substrate concentrations.

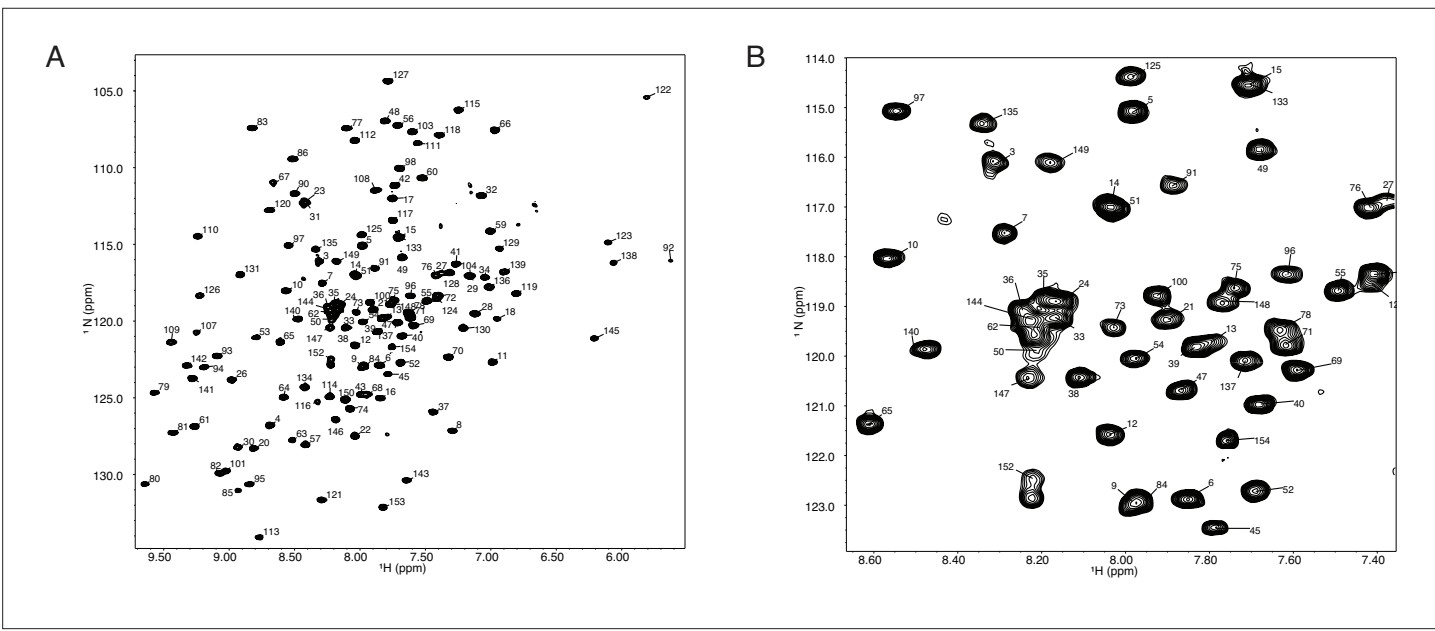

**Figure 7.** (**A**) (¹H,¹⁵N)-TROSY spectrum of 150 mM Tae1 in 20 mM HEPES, 100 mM NaCl, pH 7.0 showing resonance assignments for backbone amides. (**B**) Expanded view of a smaller region of the same spectrum.

The online version of this article includes the following source data for figure 7:

**Source data 1.** Tae1 C30A peak assignments from NMR experiments.

**Source data 2.** Tae1 C30A chemical shift perturbations from NMR experiments.

## NMR experiments

NMR experiments for backbone assignments were performed at 25°C on a Bruker 950 MHz AVANCE (Institut de Biologie Structurale in Grenoble) or Varian INOVA spectrometers (600 and 800 MHz instruments at Pacific Northwest National Laboratories). NMR samples for determination of Tae1 resonance assignments consisted of 0.3–0.6 mM [$^{13}$C, $^{15}$N]-Tae1 in citrate buffer pH 5.5 with 150 mM NaCl, 100 μM EDTA containing 5% D$_2$O. Assignment of backbone resonances was accomplished by analysis of standard triple-resonance experiments ($^1$H, $^{15}$N)-HSQC, HNCA, HNCOCA, HNCACB, CBCA(CO) NH, and HNCO spectra (*Sattler, 1999*). Assignments of Tae1 resonances in 20 mM HEPES or NaP$_i$,150 mM NaCl, 100 μM EDTA at pH 7.0 were determined by collecting a series of [$^1$H, $^{15}$N]-HSQC spectra titrating the sample from pH 5.5–7.0. These assignments were confirmed by collecting an HNCO spectrum under the final conditions at pH 7.0. NMR data were processed and analyzed using NMRPipe (*Delaglio et al., 1995*) and NMRView (*Johnson and Blevins, 1994*; *Downing, 2004*) (for the residue assignments, see *Figure 7*). See *Figure 7—source data 1* for the peak values and *Figure 7—source data 2* for CSP (Δδ) values.

All titration experiments were conducted at the Institut de Biologie Structurale in Grenoble or at the UCSF NMR facility in Genentech Hall (San Francisco). PG ligand was dialyzed against water using a float dialysis membrane device with a 100–500 Da cutoff (Spectrum Laboratories, Inc) and lyophilized before preparation of a concentrated stock solution in NMR buffer. Before aliquoting the samples or before lyophilization, the concentration of the stock solution was estimated by liquid-state NMR using the Eretic pulse sequence (*Frank et al., 2014*) and a reference 1 mM sucrose sample in 90%:10% H$_2$O:D$_2$O. Interaction studies with $^{15}$N-labeled Tae1 and PG substrates were monitored by superimposition of $^{15}$N-BEST-TROSY spectra at 25°C for different substrate-to-protein ratio (*Favier and Brutscher, 2011*). A 50 μM $^{15}$N-labeled Tae1$^{C30A}$ was titrated with 500 and 1000 μM of purified muropeptide. CcpNmr2.4 was used to monitor protein CSP for every assigned amide resonance by superimposition of the $^{15}$N-BEST-TROSY spectra. CSPs (Δδ) were calculated on a per-residue basis for the highest substrate-to-protein ratio.

## *E. coli* lysis assays

The pBAD vector was used to clone Tae1 variants of interest (*Guzman et al., 1995*). The Tae1 wild-type was initially amplified from a previously published pET22::Tae1 construct (*Chou et al., 2012*), together with the pelB signal sequence. All variants were constructed based on this pelB_pBAD::Tae1 template. See *Table 2* for all the primers used to clone Tae1 and the different variants. SOE-PCR was used to introduce the individual mutations, followed by restriction enzyme digestion and ligation. pBAD::Tae1 constructs were always transformed freshly into *E. coli* BW25113 chemically competent cells and an LB culture with antibiotic was immediately started after the transformation outgrowth period. The culture was diluted 1–100 on the following day into LB media in the presence of antibiotic selection. After reaching OD$_{600}$ of 0.2, protein expression was induced with arabinose (0.25% in *Figure 3* and 0.0025% in *Figure 4* and *Figure 5*) and cell lysis was monitored overnight on a BioTek plate reader. See *Figure 3—source data 1*, *Figure 4—source data 1*, and *Figure 5—source data 1* for full blot images.

## DMS library construction and assay

pET22b vector was chosen for library construction. This vector contains the pelB signal sequence that would allow the expression of Tae1 variants in the cell periplasm. The preparation of variants was achieved through an error-prone PCR mutagenesis kit (GeneMorph II Random Mutagenesis Kit, Agilent Technologies), following the manufacturer's instructions in order to ensure high proportion of single-mutant variants. The variants were cloned via restriction enzyme cloning. In order to track the behavior of variants, each one was barcoded with a unique DNA sequence. The barcodes were introduced immediately after the Tae1 coding sequence via Gibson assembly cloning (In-Fusion HD Cloning Kit, Takara Bio). Our next-generation sequencing strategy comprised two stages. First, the entire library was sequenced to determine the barcode corresponding to each variant. This was accomplished using 300 bp forward and reverse reads on the MiSeq Illumina platform (MiSeq Reagent Kit v3). Using this information, a dictionary was prepared to demonstrate the relationships between variants and barcodes. The dictionary was prepared by following previously published scripts (*Mavor et al., 2016*) with some modification (see the GitHub directory 'ChouLab dictionary' for the steps and

**Table 2.** X-ray data collection and refinement statistics for Tae1[C110S] (PDB ID: 7TVH).

| | Crystal 1 |
|---|---|
| **Data collection** | |
| Space group | $P\ 1\ 21\ 1$ |
| Cell dimensions | |
| $a, b, c$ (Å) | 39.121 108.512 82.402 |
| Resolution (Å) | 65.56–1.71 (1.77–1.71) |
| $R_{sym}$ or $R_{merge}$ | 0.03896 (0.9019) |
| $I / I$ | 10.48 (0.80) |
| Completeness (%) | 98.45 (87.02) |
| Redundancy | 2.00 (2.00) |
| $CC_{1/2}$ | 0.999 (0.469) |
| | |
| **Refinement** | |
| Resolution (Å) | 65.56–1.71 |
| No. reflections | 72,921 (6436) |
| $R_{work}/R_{free}$ | 0.2113/0.2430 |
| No. atoms | 4,673 |
| Protein | 4,277 |
| Water | 396 |
| $B$-factors | |
| Protein | 38.21 |
| Water | 43.00 |
| r.m.s. deviations | |
| Bond lengths (Å) | 0.006 |
| Bond angles | 0.77 |
| Ramachandran favored (%) | 98.09 |
| Ramachandran allowed (%) | 1.91 |
| Ramachandran outliers (%) | 0 |
| Rotamer outliers | 0 |

Values in parentheses are for the highest-resolution shell.

scripts). For each of the subsequent DMS experiments, 25 bp reads on a NextSeq Illumina platform (conducted at Chan Zuckerberg BioHub) were used to sequence only the barcode region. For data analysis and DMS scores calculations, we used Enrich2 software (*Rubin et al., 2017*). The barcode dictionary was used to convert barcodes to variants. Please see the 'Choulab DMS data analysis' GitHub directory for detailed information on data analysis in Enrich2. For the DMS experiments in this study, the DMS library was grown for 3 hr at 37°C in M9 minimal medium (or M9 supplemented with 20 mM D-Met) before distributing across a 96-well plate. A set of 24 wells was combined for each experimental time point in order to obtain enough DNA for library amplification. Libraries were amplified for nine cycles using Phusion high-fidelity polymerase to minimize any masking of differences in abundance. The libraries were gel purified, quantified using Qubit, and their concentrations were adjusted according to Illumina instructions before sequencing. The 0 hr time point (DMS library before any growth) was chosen as the input of the 96-well plate incubation. After growth to $OD_{600}$ 0.2, protein expression was induced with 1 mM IPTG. Cells were collected 2 hr after induction that would serve as the output time point. All of the scripts used in the article are available on GitHub at

## Clustering analysis

K-means clustering analysis was conducted using previously published scripts (**Thompson et al., 2020**) to group Tae1 positions into categories based on their fitness scores. We calculated spatial clusters by (1) sorting the fitness score vectors for each position, (2) trimming the vectors to match vector lengths (discarding no-data values), (3) calculating a difference by subtracting the two sorted and trimmed vectors, and (4) calculating the distance – mean of the absolute value of the vector difference. The distance between a candidate position and a cluster of positions is calculated as the average of the distance between the candidate position and the three closest nonself positions in the cluster. Clustering was performed over 10 rounds following the initial seeded round, and convergence was confirmed by observing that five repetitions gave identical clusters. The results yielded five clusters, including an LOF phenotype cluster, which was the focus of the study. The same analysis was performed for both DMS experiments. See the GitHub directory 'ChouLab clustering' for the steps and scripts. All of the scripts used in the article are available on GitHub at https://github.com/AtanasDRadkov/ChouLab_DMS (copy archived at swh:1:rev:e07e30f99f36b3ad90f042c50e2f-4b3c0241e7d4) (**Radkov, 2022**).

## HPLC PG analysis

PG isolation and HPLC analysis were conducted as described (**Stankeviciute et al., 2019**) with the following modifications. Briefly, *E. coli* BW25113 cells were grown in 200 ml LB (or LB plus 20 mM D-Met) to mid-log phase. The cells were harvested by centrifugation at 5000 × *g* for 15 min, after which they were suspended in 300 µl fresh LB and added to 600 µl of 6% SDS solution in a 2 ml Eppendorf tube. The cells were boiled for 3 hr in a heat block and then maintained in the SDS solution at 4°C until needed. Before any treatments, the boiled sacculi were washed five times with MQ water to remove the SDS. Each wash consisted of 15 min centrifugation at 21,000 × *g* followed by decanting of the supernatant and resuspending in 1 ml MQ water by vortexing. After the final wash, the sacculi were suspended in Pronase E digestion buffer (10 mM Tris-HCl pH 7.2% and 0.06% NaCl) and were treated for 2 hr at 60°C with Pronase E (100 µg/ml final concentration, pretreated for 2 hr at 60°C in the digestion buffer; supplier: VWR). The sacculi were incubated at 100°C for 10 min to inactivate the Pronase E. Enzymatic treatments with Tae1 variant enzymes were conducted at this point using 20 mM HEPES pH 7.0, 100 mM NaCl buffer. Tae1 enzyme was added to 1 µM final concentration, and the reactions were incubated at room temperature for 10 min, followed by 10 min at 100°C to inactivate the enzyme. The sacculi were washed three times as described above. Finally, the sacculi were solubilized using 25 µg/ml final concentration of mutanolysin (Sigma-Aldrich) in 10 mM Tris-HCl pH 7.2% and 0.06% NaCl after overnight digestion in a 37°C shaker. To prepare PG fragments for HPLC analysis, the mutanolysin treatment was incubated at 100°C for 10 min, followed by centrifugation to remove any debris, alkalinization with 0.5 M borate buffer and reduction with sodium borohydride for 1 hr at room temperature. The reduced PG fragments were acidified to pH 3–4 with 50% o-phosphoric acid and filtered (PVDF syringe filters, 0.22 µm, 0.4 mm, Tisch Scientific) before the HPLC injection. The HPLC instrument, method, and buffers were the same as described previously (**Hayes et al., 2020**).

## Protein purification

All Tae proteins, and Tai1 protein, used in this study were purified according to an established protocol (**Chou et al., 2012**) using pET29b+ vector and 3 hr induction in a 37°C shaker with 1 mM IPTG. Buffers for metal affinity chromatography were 20 mM HEPES pH 7, 100 mM NaCl, 20 mM imidazole and 20 mM HEPES pH 7, 100 mM NaCl, 400 mM imidazole. Buffer for SEC was 20 mM HEPES pH 7, 100 mM NaCl. The SEC was performed using an isocratic gradient on an AKTA pure instrument with a HiPrep 16/60 Sephacryl S-200 HR column. For isotopically labeled Tae1[C30A], the same plasmid constructs were used, but the expression was conducted in M9 minimal medium prepared with [15]N-ammonium chloride (5 g per liter media) (M9 minimal medium prepared according to Cold Spring Harbor Protocols). The cells were grown and induced as described above. The same purification procedure was used as described above.

## Phylogenetic trees

Initial candidates of the Tae1 and Tai1 families as identified in *Russell et al., 2012* were aligned and used as a seed in a PSI-BLAST search. Significant results were downloaded and aligned using MAFFT. Candidates containing the critical Cys-His dyad were kept, and other candidates discarded. Final candidates were realigned. Default parameters were used for all steps. Multisequence alignment (ClustalW algorithm) and phylogenetic trees (PhyML analysis) were prepared on GenomeNet (at https://www.genome.jp/). See *Figure 5—source data 2* for all Tae1 proteins and *Figure 6—figure supplement 2—source data 1* for the Tai1 proteins.

## *P. aeruginosa* strain construction

*P. aeruginosa* strains used in this study were derived from the sequenced strain PAO1 (*Stover et al., 2000*). *E. coli* strain SM10 was used for conjugal transfer of pEXg2 plasmids for homologous recombination as described previously (*Russell et al., 2011*). All deletions were in-frame and were generated by exchange with deletion alleles constructed by SOE PCR, followed by restriction enzyme cloning into pEXg2 vector. See *Table 1* for the primers used to delete genes *pppA* and *retS*, and to introduce the C110S mutation into *tae1*.

## Contact-dependent fitness assays

Fitness assays were performed with overnight *P. aeruginosa* cultures. The assays were performed on solid media (LB without added NaCl containing 3% agar) using nitrocellulose membranes (Amersham Biosciences Hybond-N+, 0.45 μm) to ensure close contact between the cells (*Russell et al., 2011*). The $OD_{600}$ was adjusted to 1.0 for all strains. 10 μl were spotted onto a membrane to initiate the contact-dependent fitness assays. The assays were incubated for 6 hr at 37°C after which the spots were removed with a razor blade, placed in a tube with glass beads, shaken on a bead beater (BeadBug microtube homogenizer; Z763705, Sigma) for 20 s at 4000 rpm, and the suspended cells were distributed across a 96-well plate (3 × 200 μl wells for each competition tube) for $OD_{600}$ measurements on a BioTek plate reader. See *Figure 6—source data 1* for full blot images.

## Contact-dependent competition assays

Competition assays were performed with overnight *P. aeruginosa* cultures. The assays were performed on solid media (LB without added NaCl containing 3% agar) using nitrocellulose membranes (Amersham Biosciences Hybond-N+, 0.45 μm) to ensure close contact between the cells (*Russell et al., 2011*). The $OD_{600}$ was adjusted to 1.0 for all strains. 50 μl of each *P. aeruginosa* strain was mixed with 50 μl of *E. coli*. We used an *E. coli* BW25113 strain expressing a genomic copy of a red fluorescent protein (inserted at the *nfsA* gene by lambda red recombineering [*Silvis et al., 2021*]). 10 μl were spotted onto a membrane to initiate the contact-dependent competition assays. The assays were incubated for 6 hr at 37°C after which the spots were removed with a razor blade, placed in a tube with glass beads, shaken on a bead beater (BeadBug microtube homogenizer; Z763705, Sigma) for 20 s at 4000 rpm. To obtain cell counts at the initial time point of the assay, 1% paraformaldehyde was used to fix 100 μl of *P. aeruginosa* and *E. coli* mixed cells for 20 min at room temperature. The reaction was quenched with 0.5 M glycine for 5 min at room temperature. The same fixation step was performed with 100 μl of cells extracted from each nitrocellulose membrane spot at the end of the competition. The cell counts were obtained by injecting 20 μl of fixed cells onto a flow cytometer (Attune NxT by Thermo Fisher), using three mix cycles, a flow rate of 100 μl per min, BL1 (340 nm) and YL2 (360 nm) lasers, and forward scatter at 900 and side scatter at 500. The competitive index represents the ratio of the final cell counts over the initial cell counts. The cell counts at each time point were calculated as the number of *P. aeruginosa* cells divided by the number of *E. coli* cells. Flow cytometer data were analyzed using the FlowJo software. See *Figure 6—source data 1* for full blot images.

## Isothermal titration calorimetry

Isothermal titration calorimetry experiments were performed at 298 K using a Microcal ITC200 (GE Healthcare, Northampton, MA). Stock solutions of Tae1 and Tai1 were dialyzed overnight at 277 K in the same buffer (50 mM NaCl, 20 mM Tris 7.0). The cell was filled with 200 μl of Tai1 protein at a concentration of 20 μM to which 150 μM of Tae1 protein was titrated. For 25 titrations of 1.5 μl of

Tae1 into Tai1, the heat of binding was recorded. Data were extracted and fitted using the single-site binding model following the manufacturer's instructions.

## Tae1$^{C110S}$ crystal structure

Tae1$^{C110S}$ protein crystals were generated by hanging drop vapor diffusion at 25°C from a 1:1 mixture of 10 mg/ml protein (in 20 mM HEPES pH 7, 100 mM NaCl, 5% glycerol) with 0.2 M sodium thiocyanate, 0.1 M HEPES pH 7.5, 20% PEG3350 for 24 hr (same conditions that were used for initial structure of Tae1 [*LaCourse et al., 2018*]). Crystals were directly used for diffraction data collection at the Lawrence Berkeley National Laboratory Advanced Light Source Beamline 8.3.1 (*MacDowell et al., 2004*). Coot (*Emsley and Cowtan, 2004*), and maximum-likelihood (*Adams et al., 2010*) refinement with PHENIX were used for iterative building and refinement. Structure was validated by MOLProbity (*Davis et al., 2004*) and visualized with Chimera (*Pettersen et al., 2004*). Coordinates and structure factors were deposited in the Protein Data Bank (PDB ID: 7TVH). See *Figure 6—source data 1* for validation report, as well as the maps and coordinates of the deposited structure.

## Sacculi pull-down assays

Sacculi were isolated as described above from 500 ml cell cultures, either with or without D-Met. The sacculi from each media were used in eight pull-down assays, four assays using 1 μM enzyme and four assays using 0.1 μM enzyme. In each assay, the E-64 inhibitor (MilliporeSigma; Cat# E3132) was added at 10-fold excess of the enzyme concentration. The assays containing enzyme, inhibitor, and sacculi were incubated for 1 hr at 37°C with shaking, followed by centrifugation to pellet the sacculi, completely removing the supernatant, resuspending the sacculi in 2× Laemmli sample loading buffer, and boiling for 15 min at 100°C in preparation for SDS-PAGE analysis and Western blotting. See *Figure 4—figure supplement 2—source data 1* for full blot images.

## Acknowledgements

We thank Dr Doug Fowler (University of Washington) and Dr Eric Chow (University of California – San Francisco) for their help with setting up the deep mutational scanning pipeline. We are also grateful to Dr David Mavor (University of Massachusetts Chan Medical School), Dr Robert Newberry (University of Texas – Austin), Dr Martin Kampmann (University of California – San Francisco), and Dr James Fraser (University of California – San Francisco) for their assistance with sequencing data analysis. We thank Dr Kari Herrington and Dr SoYeon Kim (University of California – San Francisco) for their contribution with microscopy data collection and analysis. We also thank Dr George Meigs and Dr James Holton (Lawrence Berkeley National Lab), and Dr Michael Thompson (University of California – Merced), for their help with X-ray crystallography data collection and analysis. We are grateful to Dr Mark Kelly for his help with setting up NMR data collection (University of California – San Francisco). We also thank Dr Parnian Lak (University of California – San Francisco) for her help with setting up the ITC experiments. Lastly, we greatly appreciate the input regarding the experiments and the manuscript from members of the Chou lab at the University of California – San Francisco (Kristine Trotta, Calla Martyn, Fauna Yarza, Elizabeth A Martinez-Bond, Patrick Grimes, Dr Domokos Lauko). AS received a Life Sciences Research Foundation fellowship from the SVCF-Wave Fund. Financial support from the IR-RMN-THC FR3050 CNRS for conducting the research on the 950 MHz NMR spectrometer is gratefully acknowledged. We would also like to thank the Chan Zuckerberg Biohub as well as the Sanghvi-Agarwal Innovation Award for their financial support.

## Additional information

### Competing interests

Seemay Chou: currently the President and CEO of Arcadia Science, a for-profit research company focused on non-model organisms. While work presented in this manuscript is not related and will not be continued at Arcadia, I thought it would be prudent to declare this position which may be perceived by some as a competing interest. The other authors declare that no competing interests exist.

## Funding

| Funder | Grant reference number | Author |
|--------|------------------------|--------|
| Centre National de la Recherche Scientifique | IR-RMN-THC FR3050 | Jean-Pierre Simorre |
| Chan Zuckerberg Biohub | | Seemay Chou |
| University of California, San Francisco | Sanghvi-Agarwal Innovation Award | Seemay Chou |
| Life Sciences Research Foundation | | Anne L Sapiro |

The funders had no role in study design, data collection and interpretation, or the decision to submit the work for publication.

## Author contributions

Atanas Radkov, Conceptualization, Data curation, Formal analysis, Investigation, Methodology, Writing - original draft; Anne L Sapiro, Formal analysis, Investigation, Validation, Visualization, Writing – review and editing; Sebastian Flores, Hayden Saunders, Rachel Kim, Steven Massa, Chase Mateusiak, Jacob Biboy, Ziyi Zhao, Investigation; Corey Henderson, Formal analysis, Investigation; Samuel Thompson, Formal analysis; Lea M Starita, Conceptualization; William L Hatleberg, Writing – review and editing; Waldemar Vollmer, Conceptualization, Project administration, Writing – review and editing; Alistair B Russell, Formal analysis, Project administration, Writing – review and editing; Jean-Pierre Simorre, Peter Brzovic, Conceptualization, Investigation, Project administration, Writing – review and editing; Spencer Anthony-Cahill, Conceptualization, Formal analysis, Investigation, Writing – review and editing; Beth Hayes, Conceptualization, Investigation, Project administration, Writing - original draft, Writing – review and editing; Seemay Chou, Conceptualization, Formal analysis, Funding acquisition, Project administration, Supervision, Writing – review and editing

## Author ORCIDs

Atanas Radkov (ID) http://orcid.org/0000-0001-7418-279X
Anne L Sapiro (ID) http://orcid.org/0000-0002-6612-8272
Hayden Saunders (ID) http://orcid.org/0000-0002-7582-3031
Rachel Kim (ID) http://orcid.org/0000-0003-3793-4264
Chase Mateusiak (ID) http://orcid.org/0000-0002-2890-4242
Jacob Biboy (ID) http://orcid.org/0000-0002-1286-6851
William L Hatleberg (ID) http://orcid.org/0000-0002-0423-7123
Jean-Pierre Simorre (ID) http://orcid.org/0000-0002-7943-1342
Beth Hayes (ID) http://orcid.org/0000-0001-6633-751X
Seemay Chou (ID) http://orcid.org/0000-0002-7271-303X

## Decision letter and Author response

Decision letter https://doi.org/10.7554/eLife.79796.sa1
Author response https://doi.org/10.7554/eLife.79796.sa2

---

# Additional files

## Supplementary files

• MDAR checklist

## Data availability

-Deep mutational scanning dataset has been deposited to NCBI, Sequence Read Archive - identifier: PRJNA803461 -X-ray structure crystallography dataset has been deposited to PDB - ID: 7TVH -NMR resonance peak assignments have been directly provided in the manuscript -Custom scripts for DMS analyses have been uploaded at GitHub at https://github.com/AtanasDRadkov/ChouLab_DMS (Copy archived at swh:1:rev:e07e30f99f36b3ad90f042c50e2f4b3c0241e7d4) -All other data generated or analyzed in this study are included in the manuscript and supporting files.

The following datasets were generated:

| Author(s) | Year | Dataset title | Dataset URL | Database and Identifier |
|---|---|---|---|---|
| Radkov AD | 2022 | T6S amidase effector 1 deep mutational scanning | https://www.ncbi.nlm.nih.gov/bioproject/PRJNA803461 | NCBI BioProject, PRJNA803461 |
| Radkov A, Saunders H, Chou S | 2022 | Hyperlytic variant of Tae1, Type VI secretion amidase effector 1, from *Pseudomonas aeruginosa* (Cys110Ser) | https://www.rcsb.org/structure/unreleased/7TVH | RCSB Protein Data Bank, 7TVH |

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
