## [Editor Report]

This study investigates the factors underlying differences in the antimicrobial efficacy of members of the T6SS amidase effector (Tae) superfamily of toxins. This is an interesting and important question from both a physiological and evolutionary perspective.

---

## [Decision Letter]

[Editors' note: this paper was reviewed by Review Commons.]

---

## [Author Response]

General Statements

In our manuscript, we present a multi-pronged approach to examine the genetic and molecular determinants of an antimicrobial cell wall-degrading toxin found in bacteria. We focused our work on the lytic activity of a toxin known as Tae1 involved in Type VI Secretion-mediated competition between bacteria.

All three reviewers provided valuable and actionable feedback on our manuscript. We were particularly pleased to see that many of the reviewer points were common across more than one reviewer. Of particular note, reviewers requested (1) improvement in technical quality for several Western blot analyses and (2) one more layer of experimental evaluation regarding how different bacterial cell wall structures may impact toxin variant binding in vitro. We are confident that we can experimentally address these two major suggestions in a timely manner, as described in Section 2. Other more minor points of feedback and our responses are also described in this section.

A third major suggestion from one of the reviewers is to more fully characterize the effects of some Tae1 variants on in vivo interbacterial competition assays. For reasons we elaborate on in Section 4, we feel that the nature of this experiment falls outside the scope of this study. Furthermore, the experiments required to pursue this direction rigorously fall outside the scope of what is feasible for us at this time.

Description of the planned revision (1 total) - completed

*Request.* All 3 reviewers asked us to provide more experimental evidence addressing the hypothesis that differential PG affinity across Tae1 variants could explain variation in toxic activity.

*Response.* We agree that this is an interesting point to follow up on further. To be clear, we also do not know whether this hypothesis is true at this stage, but we have devised an approach to ask the question experimentally across a subset of our DMS variants.

*Experimental plan.* Reviewer 1 suggested that we quantify in vitro binding affinities for PG using isothermal titration calorimetry (ITC). However, given that ITC requires high concentrations of well-defined homogeneous substrates, which we are not able to generate for more complex higher order structures of cell wall PG, we propose a pull-down based approach.

To summarize, we propose conducting pull-downs using insoluble, purified cell wall sacculi from our two *E. coli* grown under the two conditions as bait for recombinant Tae1 proteins. Given that intact sacculi or inherently insoluble, we can simply collect bound Tae1 through centrifugation of sacculi pellets and examine the amount of Tae1 associated by Western blot analysis. These will need to be conducted across a titration of Tae1 concentrations and also with catalytic activity inhibited to avoid solubilization of sacculi. We will attempt this through either C30A backgrounds or in the presence of a chemical or Tai1 inhibitor.

If there is indeed differential affinity for PG underlying Tae1 variant functional differences, we would expect to see greater relative association of Tae1 variants with the type of cell wall sacculi that they more effectively lyse in our DMS screen. We would expect the reverse trend to also be true (lower affinity for less active variants).

*Result*. While we were waiting to hear from *eLife,* we completed this proposed experiment above. In summary, we were unable to detect any major differences in Tae pelleting across conditions (see new Figure S5). Although the sensitivity of our pull-down assay inherently limits are detection to gross differences, our results suggest that there are no dramatic differences in bulk sacculus binding properties across conditions or variants.

*Experimental revisions (2 total)*
– Improve technical quality of Western blot analyses (Reviewers 2, 3)

Two reviewers commented on the smeared appearance of Tae1 bands in our Western blot analyses. We repeated these experiments with more careful attention to lysate preparation, using a higher percentage SDS gel for better separation of low molecular weight proteins as suggested. Although we still saw some “smeariness” due to the fact that these cells have undergone lysis and tend to be “gooey”, these modifications did indeed improve the crispness of our bands. See updated panels for Figure 4F and 5B.

– Examine expression levels of WT and C110S Tae1 variant in vivo (Reviewer 2)

Reviewer 2 requested additional context regarding Tae1 expression levels in P. aeruginosa to more precisely assess whether increased self-intoxication by Tae1^C110S^ in P. aeruginosa was due to differences in toxin activity or toxin levels. This was a good suggestion, and we examined this in two ways by Western blot analysis: comparison of native Tae1 expression levels in P. aeruginosa cells (1) grown alone in liquid culture, and (2) in P. aeruginosa cells grown together with *E. coli* on solid media under conditions matching our competition assays in this study. Our analyses suggest that Tae1^C110S^ does not express at a higher level than Tae1^WT^, ruling out the possibility that variation in bulk expression levels underlie competition outcomes we observe.

1. Other questions or text-based revisions

All the requests below can be readily addressed in our revision.

– Reviewer 1

*Correct typo figure 5 legend (“does is not”)*

*Correct citation in line 108 – “that does not seem to support a physiologically relevant context for Tae”*

*Line 122/123 – something missing in the sentence*

*Line 148 – clarify if we sequenced plasmid barcodes or the mutated ORFs*
Description of analyses that authors prefer not to carry out (2 total)

Below we describe two sets of experiments that we prefer to not carry out. While we actually agree that results for these suggested lines of inquiry would indeed be interesting (we hope to see others do it!), we feel that these analyses fall outside the scope of this manuscript. Given the labor intensiveness of both requests, we also unfortunately must comment that these are not feasible for us to execute rigorously. Please see more information about each of these aspects below.

1. Analysis of additional DMS variants in interbacterial competition assays (Reviewer 2)

*Scope.* We absolutely agree that examining Tae1 variants in the context of interbacterial competitions would be a critical orthogonal approach in order to validate that the DMS results have any bearing on competition outcomes. However, we feel that major focus of this paper is on the more molecular and biophysical insights that our approach can offer. Our study tests our assumptions about the kinds of features and surfaces that are important for proteins that engage with non-canonical complex substrates. It is, of course, interesting to think about the implications of this for physiological phenotypes and the drivers of toxin evolution. It is also exciting to imagine how this kind of information could be used to one day engineer certain interbacterial outcomes. We hope that others in the field will push our efforts into these directions, but we do not feel that these directions are essential for our conclusions. However, our conclusions on the molecular and biophysical aspects have helped generate interesting hypotheses in microbial ecology that could be followed up on by others.

*Feasibility*. In order to conduct well-controlled *P. aeruginosa: E. coli* competition assays for our Tae1 variants, we would need to generate a significant number of new *P. aeruginosa* strains encoding point mutations for each of our variants across several genetic backgrounds. The competitions themselves also require a considerable amount of work to optimize and quantify. The first author of this paper, who was the primary driver of this work, is no longer in my lab or in academia. As for myself, I am also in the middle of a transition out of academia and am actively ramping down my lab at UCSF. I no longer have the space or appropriate set-up to support this longer-term effort.

1. Conduct additional DMS screens in more conditions (Reviewer 1)

*Scope.* We love this idea! In fact, we hope that others are motivated to adopt our workflow to run many more DMS screens for T6S toxins, as we believe these screens provide a lot of useful and sometimes surprising insights that could be of great interest to others. However, we believe that the primary goal of this paper is to establish this methodology as a compelling approach for studying toxins and, more generally, proteins with complex cellular substrates. It does not

necessarily fall within the scope of this paper to fully assess the mechanistic implications of cell wall diversity across a wide range of conditions.

*Feasibility.* In our experience, rigorously conducting DMS screens requires a significant amount of effort and resources to establish consistent experimental conditions. Also, a non-trivial number of costly sequencing-based experiments are required across control and variables for the results to be statistically sound and meaningful. Furthermore, experimental validation of results are ultimately important for our ability to confidently generate hypotheses stemming from these datasets. As stated above, the first author of this paper, who was the primary driver of this work, is no longer in my lab or in academia. As for myself, I am in the middle of a transition out of academia and am actively ramping down my lab at UCSF. I no longer have the space or appropriate set-up to support this longer-term effort.